# Imaging translational control by Argonaute with single-molecule resolution in live cells

Charlotte A. Cialek[1], Gabriel Galindo[1], Tatsuya Morisaki [1], Ning Zhao[1], Taiowa A. Montgomery [2✉] & Timothy J. Stasevich [1,3✉]

A major challenge to our understanding of translational control has been deconvolving the individual impact specific regulatory factors have on the complex dynamics of mRNA translation. MicroRNAs (miRNAs), for example, guide Argonaute and associated proteins to target mRNAs, where they direct gene silencing in multiple ways that are not well understood. To better deconvolve these dynamics, we have developed technology to directly visualize and quantify the impact of human Argonaute2 (Ago2) on the translation and subcellular localization of individual reporter mRNAs in living cells. We show that our combined translation and Ago2 tethering sensor reflects endogenous miRNA-mediated gene silencing. Using the sensor, we find that Ago2 association leads to progressive silencing of translation at individual mRNA. Silencing was occasionally interrupted by brief bursts of translational activity and took 3–4 times longer than a single round of translation, consistent with a gradual increase in the inhibition of translation initiation. At later time points, Ago2-tethered mRNAs cluster and coalesce with P-bodies, where a translationally silent state is maintained. These results provide a framework for exploring miRNA-mediated gene regulation in live cells at the single-molecule level. Furthermore, our tethering-based, single-molecule reporter system will likely have wide-ranging application in studying RNA-protein interactions.

[1] Department of Biochemistry & Molecular Biology, Colorado State University, Fort Collins, CO 80523, USA. [2] Department of Biology, Colorado State University, Fort Collins, CO 80523, USA. [3] Cell Biology Center and World Research Hub Initiative, Tokyo Institute of Technology, Yokohama, Japan. ✉email: tai.montgomery@colostate.edu; tim.stasevich@colostate.edu

Translation is the culmination of gene expression, whereby genetic information encoded in nucleic acids is converted into proteins. This basic process is fundamental to all life, giving cells the ability to rapidly establish and maintain diverse phenotypes in response to the environment[1,2]. A multitude of factors work in concert to control which mRNAs are translated, how much peptide product is synthesized, and when to halt erroneous translation[3,4]. These regulatory factors are activated by broad cell signaling pathways that respond to stimuli including stress, growth conditions, and development[5].

The dynamics of translational control have traditionally been studied in living cells at the bulk level (e.g. Western blots, polysome/ribosome profiling) or at the single-cell level (e.g. fluorescent/bioluminescent reporters)[6,7]. However, these assays lack spatiotemporal resolution, which has made it hard to decipher complex gene regulatory mechanisms[8]. More recently, it has become possible to explore translation dynamics in living cells at the single-molecule level[9–13]. These single-molecule technologies, which we collectively refer to as Nascent Chain Tracking (NCT), all use repeat mRNA and protein tags to brightly label and track single mRNA using one fluorophore and elongating nascent peptide chains using another fluorophore (reviewed in ref. [14]). The ability to image translation at single mRNA makes it possible to quantify individual translation events, measure ribosome initiation and elongation rates, and discern the heterogeneity in translation between mRNA. Since data from this technique are acquired on a microscope, spatial information is naturally embedded. Further, several recent advances in NCT technology have made it possible to study changes in translation at individual mRNA in response to stress[15–18], mRNA subcellular location[19], mRNA sequence composition[20–24], and more (as reviewed in ref. [25]).

While powerful, one shortcoming of NCT technology has been visualizing how specific regulatory factors impact translation. Because NCT only amplifies signals from the elongating nascent peptide chain and the mRNA reporter, it is difficult to simultaneously monitor relatively weak signals from individual regulatory factors. Sometimes this difficulty can be avoided if a regulatory factor produces a strong and consistent molecular phenotype that immediately impacts the reporter or its translation. For example, siRNA-directed cleavage of a reporter by Argonaute2 (Ago2) could be detected and quantified using NCT alone because it happened rapidly (seconds to minutes) and resulted in a strong molecular phenotype (the physical splitting of fluorescence signals)[26]. However, translation is often controlled by regulatory factors that act in more subtle ways, for example by inhibiting translation initiation, stalling elongation, or promoting mRNA decay. These modes of regulation tend to have a progressive phenotype that requires extended observation with high sensitivity. To illustrate, in addition to siRNAs, Ago2 binds miRNAs that target mRNAs through partial sequence complementarity and direct gene silencing through a distinct cleavage-independent mechanism[27]. The miRNA-induced silencing complex (miRISC) consists of a miRNA and an Argonaute protein, such as Ago2, along with additional downstream effectors that together promote translational repression and mRNA decay[28]. Because miRNA-mediated gene silencing and many other gene regulatory mechanisms are likely more gradual and variable than silencing directed by siRNAs, exploring their impact on translation at the single-molecule level is inherently more difficult.

To confront this problem, we developed a "Translation and tethering" (TnT) single-molecule biosensor that extends NCT technology. As the name implies, the TnT biosensor builds on NCT by adding a tethering cassette that stochastically recruits a fluorescently labeled regulatory factor with controllable stoichiometry.

By tracking tethered mRNA through time, the specific and direct impact and regulatory time frame of the factor can be discerned and quantified. The optimized signal-to-noise ratio of the TnT biosensor makes it possible to continually monitor in three colors single-mRNA silencing events that occur on a wide range of timescales, from seconds to hours. Thus, our TnT biosensor provides a route to do highly controlled biochemical experiments with single-molecule spatial resolution and high temporal resolution inside living cells.

Here we introduce the TnT biosensor and demonstrate its functionality by exploring miRNA-mediated gene silencing involving Ago2. We find that within minutes of tethering, Ago2 inhibits the initiation of new translation events, leading to ribosome runoff. Early occurring translational repression is likely independent of mRNA decay, as we sometimes see translation reinitiate in the presence of tethered Ago2. On longer timescales, we find evidence that tethered Ago2 interacts with endogenous miRISC machinery that accumulate at the mRNA and maintain a translationally silent state. Collectively, our data support a model in which Ago2 mediates translational repression largely through reversible inhibition of translation initiation within the cytosol followed by aggregation of mRNA-Ago2 complexes in P-bodies for sustained silencing.

## Results

**A live cell, single-molecule assay to monitor translation and protein tethering in real time.** To controllably tether factors to reporter mRNA and simultaneously visualize their impact on mRNA localization, stability, and translation, we constructed a translation and tethering (TnT) biosensor (Fig. 1A). As with standard NCT, each TnT biosensor contains 24 MS2 RNA stem loops which recruit fluorescent MS2 Coat Proteins (JF646 Halo-Tag-MCP) to label the mRNA. Translation is monitored by the localized accumulation of Cy3-conjugated α-FLAG fragmented antibodies (Fab) which can bind 10 FLAG epitopes at the N-terminus of a reporter protein. With this arrangement, as a ribosome begins to translate the TnT biosensor, the nascent protein that emerges from the ribosomal exit tunnel is rapidly labeled by Fab[9]. As multiple ribosomes engage the reporter and translation progresses, the Fab signal intensifies and colocalizes with the mRNA signal. In addition to the mRNA and nascent chain tags, the TnT biosensor also contains a tethering cassette consisting of multiple BoxB RNA stem loops adjacent to the 24 MS2 stem loops in the 3' UTR. Similar to MCP binding to MS2 stem loops, a 22-amino acid λN protein binds a 19-nucleotide BoxB stem loop with high affinity and specificity[29]. Thus, by fusing a protein of interest to λN labeled with a distinct fluorophore, it can be recruited and tethered to the BoxB stem loops within an mRNA reporter with high specificity[30,31]. We tested TnT biosensors with different numbers of BoxB stem loops. While tethering was difficult to detect with just five stem loops, with 15 stem loops we could easily detect tethering above background and track individual tethered mRNA for minutes or even hours. This allowed us to compare tethered and untethered TnT biosensors with unprecedented spatiotemporal resolution.

To demonstrate the power of the TnT biosensor, we used it to investigate the miRNA-mediated gene silencing pathway. We focused on Ago2, one of the core proteins in the pathway, because others had already shown that tethering it to an mRNA can faithfully recapitulate miRNA-mediated gene silencing[32–35]. While there is a substantial body of evidence that miRNAs regulate gene expression at the level of translation initiation, mRNA decay, and to a lesser extent translation elongation (reviewed in ref. [28]), the timing and spatial organization of these regulatory events and their relative contributions to gene silencing are unclear.

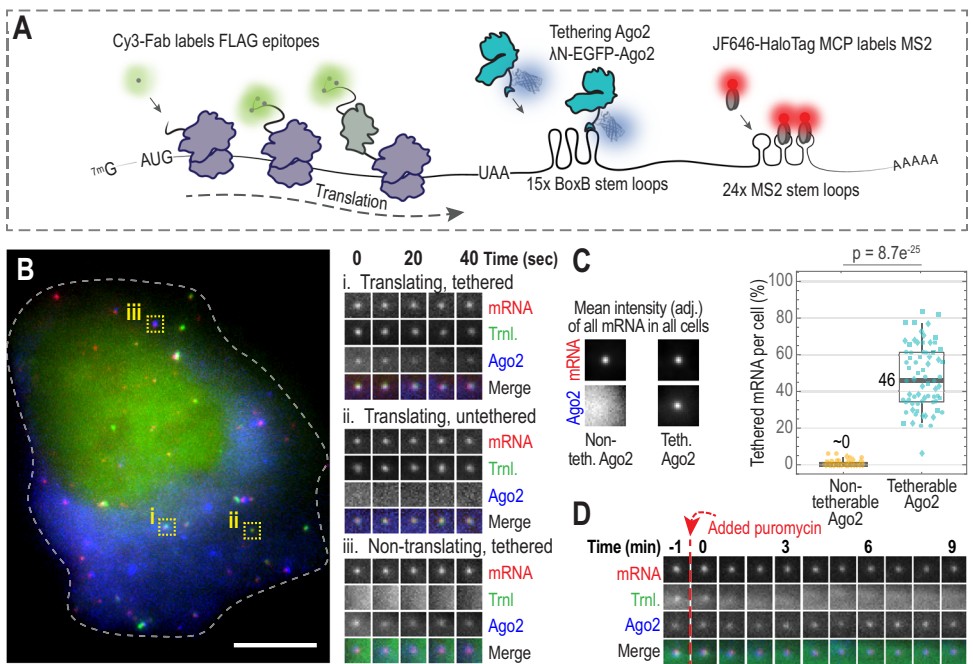

**Fig. 1 Tracking single-mRNA translation and Argonaute tethering with the TnT biosensor. A** Schematic of the Translation and Tethering (TnT) biosensor. **B** A representative cell expressing the TnT biosensor to monitor translation (Trnl.) and Ago2 tethering. Single cells (66 total) were imaged 4 h after loading plasmids encoding tetherable Ago2 (λN-EGFP-Ago2), the TnT reporter mRNA (smFLAG-KDM5B-15xBoxB-24xMS2), Cy3-FLAG-Fab, and JF646 HaloTag-MCP. The cropped images (15 × 15 pixels$^2$; 130 nm/pixel) show single mRNA across a 40 s (sec) interval: (i) translating, tethered; (ii) translating, untethered; and (iii) non-translating, tethered. The dashed line marks the cell outline. Scale bar, 10 μm. **C** The percentages of mRNAs colocalizing with tetherable (λN-EGFP-Ago2) or non-tetherable (EGFP-Ago2) Ago2 6–8 h after loading was calculated. Left, intensity-rescaled crops showing the average mRNA and Ago2 channels of all detected mRNA foci (18 × 18 pixels$^2$; 130 nm/pixel). Right, box plot showing the percentage of mRNA foci colocalizing with Ago2. Each point corresponds to a single cell (shapes denote three replicate experiments). The boxes (whiskers) show the 25–75% (5–95%) range. The *P*-value was calculated using the Mann–Whitney test, two sided. *N* = 71 (3712) and 65 (7249) cells (mRNA foci) using tetherable and non-tetherable Ago2, respectively. **D** A representative track (9 min) of a single mRNA from a cell (loaded as in B) following puromycin treatment (Time = 0, indicated by the dashed vertical red line). Crops (15 × 15 pixels$^2$; 130 nm/pixel) show the mRNA, translation, and Ago2 signal intensities. *N* = 14 mRNA in 1 cell.

To better deconvolve these complex dynamics, we engineered λN-tetherable (λN-EGFP-Ago2) and non-tetherable (EGFP-Ago2) Ago2 constructs. We then loaded the two Ago2 constructs into human U2OS cells individually along with the other TnT biosensor components (the reporter plasmid, Cy3-labeled α-FLAG Fabs to monitor translation at the reporter, and JF646 HaloTag-MCP to monitor localization of the reporter mRNA). Cells were imaged 4–6 h after introducing the TnT components. At this time, ~46% of the TnT biosensors had a detectable amount of tethered Ago2, a subset of which were translationally silent (Fig. 1B, C, Supplementary Video 1). We did not observe non-tetherable Ago2 at the TnT biosensor, indicating that the colocalization we observed was indeed due to tethering (Fig. 1C and Supplementary Fig. 1A). We next treated cells with the translational inhibitor puromycin. Consistent with the premature release of puromycylated peptide chains during active elongation[36,37], this led to a rapid loss in translation signals irrespective of whether the cells expressed tetherable or non-tetherable Ago2 (Fig. 1D, Supplementary Video 2, and Supplementary Fig. 1B, C). These data therefore demonstrate the functionality of our TnT biosensor, proving it can be used to tether a specific regulatory factor to a trackable mRNA undergoing active translation.

**Tethering Ago2 to the TnT reporter mRNA inhibits its translation.** Confident in our ability to simultaneously visualize translation and tethering at the single-mRNA level, we next explored the impact of Ago2 tethering on translation. First, we set

out to confirm that Ago2 tethering reduces total reporter protein synthesis within cells, as previously seen in bulk cell assays[32,33,35]. To control for non-specific silencing that might occur from tethering a protein to the TnT biosensor, we tethered the inert bacterial protein β-gal (λN-EGFP-β-gal), which was previously shown to have a negligible impact on translation[35]. Like Ago2, β-gal-tethered mRNAs could be actively translated and were sensitive to translation inhibition by puromycin (Supplementary Fig. 1D, E). To control for non-specific effects related to Ago2 overexpression, we also performed experiments with a non-tetherable form of Ago2 (Fig. 2A). In each experiment, we quantified total protein production from the TnT biosensor by measuring the accumulation of reporter protein KDM5B (Lysine demethylase 5B, a nuclear protein) in the nucleus over time (Fig. 2B)[9]. According to this metric, cells expressing tetherable Ago2 accumulated ~30% less KDM5B in the nucleus than cells expressing tetherable β-gal or non-tetherable Ago2 (Fig. 2C). Importantly, this result was not due to artifactual Fab accumulation in the nucleus (Supplementary Fig. 2A). As a further test, we repeated this experiment in fixed cells (looking only at a 24 h time point). Similar to live cells, protein accumulation was reduced 37–55% in fixed cells expressing tetherable Ago2 relative to cells expressing tetherable β-gal or non-tetherable Ago2 (Supplementary Fig. 2B, left). We included an additional control in this experiment in which a non-tetherable reporter mRNA lacking BoxB stem loops was coexpressed with each tetherable and non-tetherable protein construct. Protein accumulation from this non-tetherable reporter was indistinguishable between the

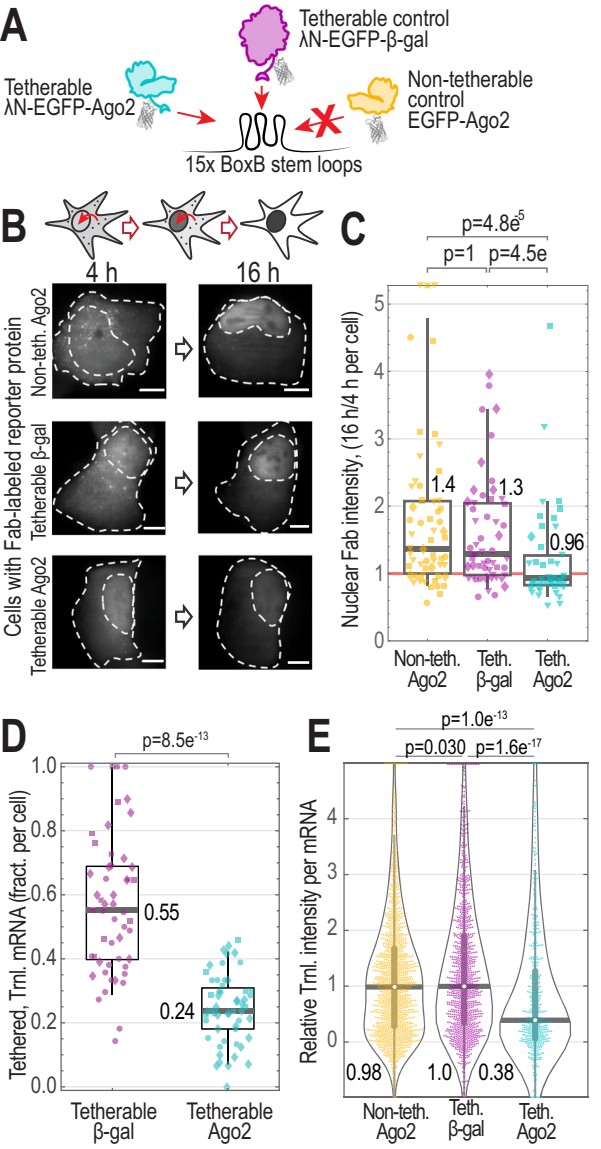

**Fig. 2 Ago2 tethering represses translation. A** Schematic of tetherable (Teth.) Ago2 and the controls: non-tetherable (Non-teth.) Ago2 and tetherable β-gal. **B** Schematic (above) and representative cells (below) showing the accumulation of the TnT reporter protein KDM5B (as marked by Fab) in the nucleus over time (4 and 16 h time points shown) after loading tetherable Ago2 or controls with the TnT components (smFLAG-KDM5B-15xBoxB-24xMS2 mRNA reporter, Cy3-FLAG-Fab, and JF646 HaloTag-MCP). Dashed lines mark the cell and nucleus (the nuclear border was defined using cytoplasmic Ago2 or β-gal staining). Scale bars, 10 μm. $N = 58$ (non-tetherable Ago2), 82 (tetherable β-gal), and 66 (tetherable Ago2) cells. **C** Box plot displaying data from B. Each data point is the ratio of KDM5B nuclear intensity at 16–4 h post-loading of the TnT components. Data from 4 replicate experiments were combined (replicates indicated by marker shape). The boxes (whiskers) show the 25–75% (5–95%) range. P values were calculated using the Mann–Whitney test, two sided. $N = 58$ (non-tetherable Ago2), 82 (tetherable β-gal), and 66 (tetherable Ago2) cells. **D** Box plot showing the fraction (fract.) per cell of Ago2- or β-gal-tethered mRNA actively translating (Trnl.) 6–8 h after TnT component loading. Data from three replicate experiments were combined (replicates indicated by marker shape). The boxes (whiskers) show the 25–75% (5–95%) range. The P-value was calculated using the Mann–Whitney test, two sided. $N = 55$ (tetherable β-gal) and 52 (tetherable Ago2) cells. **E** Violin plot displaying TnT reporter translation signal intensity in the presence of tetherable Ago2 or control constructs. The Fab signal intensity was measured for only single-mRNA foci. One representative replicate experiment is shown. P values were calculated using the Bonferroni-corrected Mann–Whitney test, two sided. $N = 18$ (1513), 28 (1158), 18 (521) cells (mRNA) for non-tetherable Ago2, tetherable β-gal, and tetherable Ago2, respectively.

the fewest mRNAs being translated (Supplementary Fig. 2C). In fact, just ~24% of Ago2-tethered mRNAs were being translated compared to ~55% of β-gal-tethered mRNAs (Fig. 2D). There was also a ~35–63% reduction in the total number of mRNA foci per cell using tetherable Ago2 compared to the controls (Supplementary Fig. 2C), indicating a shortened mRNA half-life, although we could not rule out the reduction was in part caused by mRNA clustering. Finally, individual mRNA in cells expressing tetherable Ago2 had translation signals that were 40–60% as bright as those in cells expressing tetherable β-gal (Fig. 2E and Supplementary Fig. 2D). Taken together, these data demonstrate that Ago2 tethering not only reduces the number and fraction of mRNA being translated, but also reduces the number of translating ribosomes on individual mRNAs.

**Ago2-tethered mRNA coalesce in the cytoplasm.** Due to the exceptional signal-to-noise of the TnT biosensor, we could monitor individual mRNA in living cells for extended periods of time, ranging from seconds to hours. By following mRNA in single cells over a period of 12+ h, we noticed that translationally silenced, Ago2-tethered mRNA tended to cluster over time (Fig. 3A). These mRNA clusters remained translationally silenced and colocalized with bright cytoplasmic Ago2 foci for hours. Notably, they also exhibited behaviors characteristic of phase-separation, such as coalescence (Supplementary Video 3). The average intensity of mRNA foci in cells expressing tetherable Ago2 steadily increased with time (Fig. 3B). In contrast, the average intensity of mRNA foci in cells expressing tetherable β-gal or non-tetherable Ago2 slightly decreased with time, likely due to photobleaching or probe decay (Fig. 3B).

Based on these observations, we wondered if Ago2 tethering caused the TnT biosensor to relocalize to specific subcellular locations or compartments. Ago2 is known to interact directly with TNRC6[33,38,39] and indirectly with other downstream

different protein constructs, indicating that Ago2 tethering is directly responsible for the reduction in protein output we observed with the TnT biosensor (Supplementary Fig. 2B, right). Thus, these data demonstrate that Ago2 tethering leads to a global reduction in mature protein accumulation, consistent with prior reports[32,33,35].

The main advantage of using the TnT biosensor is the ability to visualize the impact of Ago2 tethering on translation at the individual mRNA level. For this, we reimaged cells expressing both the TnT biosensor and tetherable Ago2 ~6 h after loading, when tethered and translating mRNA could easily be detected (as in Fig. 1B). To track individual mRNA for many time points, we imaged full cell volumes at a rate of 0.5 frames per second for 20 frames total. Averaging the signals from these short tracks increased the sensitivity and accuracy of translation and tethering detection. To control for tethering and Ago2 overexpression artifacts, we again imaged control cells loaded with either non-tetherable Ago2 or tetherable β-gal. For each condition, we quantified the number of mRNA that were tethered/untethered and translating/silent (Supplementary Fig. 2C). In line with the low-levels of mature reporter protein accumulation we observed in the nucleus, cells expressing tetherable Ago2 had, on average,

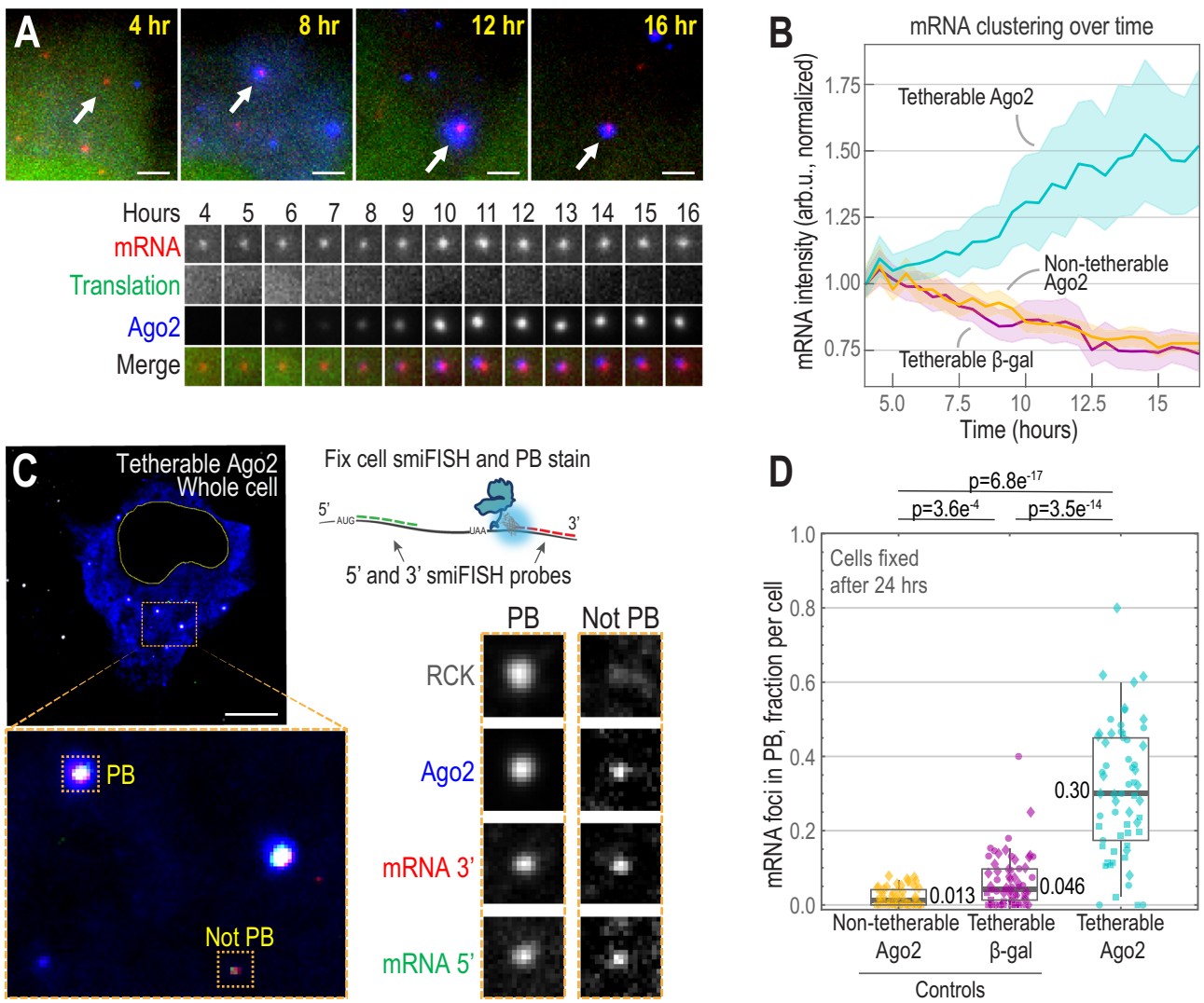

**Fig. 3 Intact Ago2-tethered TnT biosensors cluster and accumulate in P-bodies after translational silencing. A** A representative cytoplasmic patch from a live cell expressing tetherable Ago2 and the TnT biosensor and components (smFLAG-KDM5B-15xBoxB-24xMS2 mRNA reporter, Cy3-FLAG-Fab, and JF646 HaloTag-MCP) was imaged 4–16 h post loading. The mRNA identified by arrows is shown over time below in crops (15 × 15 pixels²; 130 nm/pixel). Scale bars, 2.5 μm. $N$ = 35, 46 and 44 cells for non-tetherable Ago2, tetherable β-gal, and tetherable Ago2, respectively. **B** The line plot displays mRNA clustering over time. Cells were imaged 4–16 h post loading with the TnT biosensor plus either tetherable Ago2, tetherable β-gal, or non-tetherable Ago2. The mean MCP signal (represented by the line) of all mRNA in all cells was measured over time and normalized to 1 at the first time point. $N$ = 35 (17,367), 46 (32,666) and 44 (19,236) cells (mRNA foci) for non-tetherable Ago2, tetherable β-gal, and tetherable Ago2, respectively. Shaded region, 95% CI. **C**, **D** The TnT biosensor plasmid plus either the tetherable Ago2, tetherable β-gal, or non-tetherable Ago2 plasmid were loaded into cells. After 24 h, cells were fixed. The 5' and 3' mRNA ends were labeled with smiFISH probes, and P-bodies were stained with α-RCK antibodies. **C** Representative tetherable-Ago2 cell, cytoplasmic patch, and mRNAs displaying P-body (PB) localization. Colors show P-bodies (α-RCK, gray; not shown on left), expressed tetherable Ago2 (blue), and TnT mRNA smiFISH probes (5', green; 3', red). Crops (17 × 17 pixels²; 130 nm/pixel) were autoscaled. Scale bar, 10 μm. **D** Box plots showing the fraction per cell of mRNA with 5' and 3' smFISH signals that were colocalized with P-bodies. Each point represents the fraction from a single cell (yellow, non-tetherable Ago2; purple, tetherable β-gal; cyan, tetherable Ago2; different shapes mark 1 of 3 replicate experiments). The boxes (whiskers) show the 25–75% (5–95%) range. $N$ = 56 (3,655), 63 (3,958), 61 (2,999) cells (mRNA foci) for non-tetherable Ago2, tetherable β-gal, and tetherable Ago2, respectively. $P$-values were calculated from Bonferroni-corrected Mann–Whitney tests, two sided.

effectors in the miRISC complex, including factors required for P-body assembly, such as RCK/DDX6[40–43]. Thus, we hypothesized progressive mRNA clustering could arise from multivalent interactions with endogenous miRISC machinery and/or P-bodies. Consistent with our observations of coalescing mRNAs, this machinery can exhibit phase-separation behavior both in vitro and in vivo[16,39,44,45].

To test this hypothesis, we costained fixed cells expressing the TnT biosensor with smiFISH probes[46] complementary to the 3' and 5' ends of the reporter mRNA in separate colors and with α-

RCK or α-DCP1A antibodies to label P-bodies. The bright mRNA-Ago2 foci did indeed colocalize with both P-body markers (Fig. 3C and Supplementary Fig. 3A). Furthermore, colocalization was dependent on Ago2 tethering: ~30% of mRNA colocalized with RCK in cells expressing tetherable Ago2, compared to <5% in control cells (Fig. 3D). To confirm that P-body localization was due to natural interactions with endogenous miRISC machinery, we repeated experiments using two Ago2 mutants. First, we used an Ago2 phosphomimetic mutant (Ago2$^{5XE}$) that has significantly impaired association with

mRNA targets, although it can still interact with miRNA[32]. The TnT biosensor tethered to the Ago2[5XE] mutant also localized to P-bodies, indicating that P-body localization is not facilitated by interactions between tetherable Ago2 and endogenous miRNA targets (Supplementary Fig. 3B, C). Second, we used an Ago2 mutant that lacks functional tryptophan (Trp) binding regions, making it unable to participate in diverse interactions with miRISC machinery and phase separate in vitro[39]. With this mutant, we did not observe colocalization with P-bodies (Supplementary Fig. 3D), nor did we observe any obvious impact on translation 8–12 h post loading (Supplementary Fig. 3 E). These data demonstrate that translational silencing and subsequent P-body localization of our tethering reporters is mainly driven by interactions between Trp binding regions within Ago2 and endogenous miRISC machinery.

Interestingly, the majority of mRNAs we observed contained both 5' and 3' smiFISH signals, regardless of tethering or colocalization with RCK (Supplementary Fig. 3F). While this would suggest that at least some mRNAs in P-bodies are intact, as seen previously[47], it was difficult to know for sure since each cluster contained many mRNAs and the 5' and 3' signals could therefore come from different molecules. Furthermore, since the distinct smiFISH probes bind with different affinities and have fluorophores with distinct properties, it was also difficult to say if the 5' and 3' signals were present at a one-to-one stoichiometry, although we did observe a similar ratio of signals at mRNA both inside and outside of P-bodies (Supplementary Fig. 3G).

Collectively, these data show that tethering Ago2 to an mRNA is sufficient to recruit endogenous downstream factors involved in miRNA-mediated translational repression. Furthermore, the recruitment we observed can lead to unique mRNA behaviors, including long-term translational silencing, as well as mRNA clustering and coalescence reminiscent of phase-separation. More generally, these data demonstrate how the TnT biosensor can be used to target mRNA to specific subcellular locations or microenvironments to better understand how they affect translation dynamics.

**Progressive loss of translation upon Ago2 tethering.** Thus far we have shown that Ago2-tethered TnT biosensors are translationally silenced and ultimately cluster into P-bodies. With this in mind, we next sought to zoom in on the kinetics of translation silencing by tracking single TnT biosensors prior to their clustering. According to several lines of work, Ago2 is thought to silence mRNA in part by inhibiting ribosome initiation[28]. However, direct visual evidence for this has been lacking due to the limited spatiotemporal resolution of earlier experiments. Assuming ribosome initiation is inhibited, we would expect the TnT translation signal to dim after an Ago2 tethering event. Dimming would be gradual as elongating ribosomes finish translating the open reading frame and run off the transcript one by one. Other mechanisms of translational repression are also possible and can be discerned using the TnT biosensor. For example, if ribosomes prematurely abort translation after Ago2 tethering, the translation signal would rapidly disappear, as occurs when cells are treated with puromycin (Fig. 1D, Supplementary Fig. 1B, C, E). Also, if translation elongation is repressed, ribosome progression could be slowed or halted, in which case the translation signal would persist beyond the time it takes to translate the open reading frame. Finally, if an mRNA were sliced, as in siRNA-mediated translational silencing, this would result in the physical separation of the translation and mRNA signals[26].

To distinguish between these possibilities, we developed an imaging strategy to track freely diffusing TnT biosensors with high temporal resolution for upwards of 90 min. Specifically, we imaged whole-cell volumes every 10 s in the mRNA channel, allowing us to track a single mRNA for well over an hour. Concurrently, we imaged translation and tethering once every 100 s. This staggered approach allowed us to follow the translation and tethering signals of individual mRNAs with minimal photobleaching. From 11 cells over 5 days, we identified 26 mRNA that were trackable for 35–90 min and that had at some point both tethering and translation signals. We repeated the experiment in 11 cells expressing tetherable β-gal over 4 days, finding 21 such control-tethered mRNA.

For each tracked mRNA, we measured the intensities of the mRNA, translation, and tethering signals through time. Consistent with an inhibition of translation initiation and ribosome runoff, we could observe a slow and steady decline of the translation signal from individual Ago2-tethered mRNA (Fig. 4A, Supplementary Video 4). Moreover, a scatter plot of translation signals versus tethering signals for all data points revealed a pattern: translation signals were stronger when Ago2 tethering signals were weaker, and vice versa (Fig. 4B, cyan). In contrast, this pattern (Spearman correlation coefficient $= -0.10$; $p = 1.7 \times 10^{-4}$) was not observed when mRNA were tethered to control β-gal (Fig. 4B, purple). In fact, β-gal tethering signals were strongly correlated with translation (Spearman correlation coefficient $= 0.55$; $p = 2.4 \times 10^{-81}$). This suggested that the more Ago2 present, the less likely it is that the mRNA is actively translating. To visualize this over time, we normalized all tracked signals and averaged them together (so each mRNA would have equal weight) (Fig. 4C). This revealed translation signals steadily and significantly decreased with time ($t_{1/2} \sim 40$ min), while Ago2 tethering signals steadily increased (Fig. 4C, right). The corresponding signals in control cells expressing tetherable β-gal remained steady (Fig. 4C, left). These data strongly suggest Ago2 tethering leads to a gradual runoff of ribosomes, as would be expected if ribosomal initiation were inhibited.

We next examined the individual mRNA tracks in greater detail (Fig. 4D). In line with the average behavior in Fig. 4C, the translation signals of 14/26 steadily declined as in Fig. 4A, although the rate of runoff was variable from mRNA to mRNA (Fig. 4E). This variability most likely reflected stochasticity in the start of our imaging as well as in the positioning of individual ribosomes along an mRNA upon Ago2 tethering. In particular, we noticed runoffs could be delayed by late Ago2 tethering (Fig. 4E, left) and runoffs with low initial translation signal intensities could be faster (Fig. 4E, middle vs. right), presumably because they were already in progress when imaging began. We also observed occasional bursts (6/26) of translation despite the presence of low-levels of Ago2 tethering (Fig. 4F). This would suggest the repression of translation initiation by Ago2 is reversible. Further, we observed Ago2-tethered, translationally silent mRNA (4/26) coalesce to form large clusters that were similar to the P-bodies we identified in Fig. 3 (Fig. 4G, Supplementary Video 5). These mRNA were associated with exceptionally bright Ago2 foci that likely contained many Ago2 proteins. Finally, on two occasions (2/26) we observed physical separation of the translation and mRNA signals, possibly indicating some form of mRNA cleavage[26,47] (Supplementary Fig. 4A, Supplementary Video 6). Given the scarcity of this type of event (2/26; <8% of observations), we conclude it is not a dominant form of mRNA silencing in our system.

**Translational repression at Ago2-tethered mRNA is consistent with inhibition of translation initiation.** Our analysis of the individual mRNA tracks above provides further support for a model in which Ago2 tethering strongly inhibits translation initiation. To further confirm this model, we performed Harringtonine experiments (HT). HT inhibits translation initiation

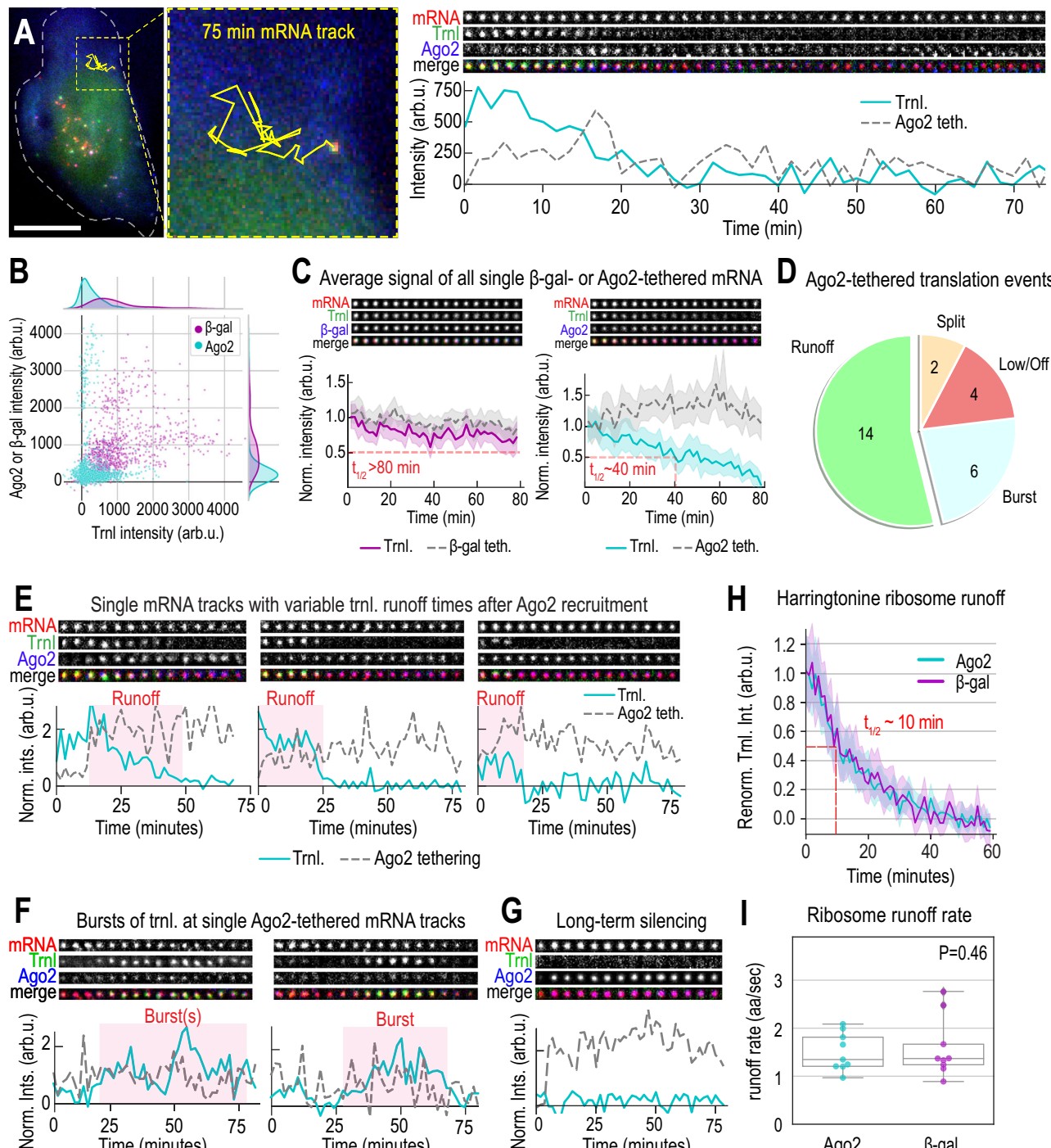

**A** 75 min mRNA track

**B**

**C** Average signal of all single β-gal- or Ago2-tethered mRNA

**D** Ago2-tethered translation events

**E** Single mRNA tracks with variable trnl. runoff times after Ago2 recruitment

**H** Harringtonine ribosome runoff

**F** Bursts of trnl. at single Ago2-tethered mRNA tracks

**G** Long-term silencing

**I** Ribosome runoff rate

while allowing already-initiated, elongating ribosomes to continue translation and run off the transcript[48]. If Ago2 tethering inhibits initiation as potently as HT, we would predict Ago2-tethered ribosome runoff times should be on the same timescale as HT-induced runoff times.

To test this prediction, we added HT to cells expressing the TnT biosensor and either tetherable Ago2 or tetherable β-gal. This led to a steady loss in the average translation signal from mRNA. Irrespective of what was tethered, the average HT-induced ribosome runoff halftime was ~10 min, with the majority of signal lost in ~20 min (Fig. 4H and Supplementary Video 7). Fits to all runoff curves from individual cells revealed the ribosomal runoff rates were statistically indistinguishable between tetherable Ago2 and β-gal, ranging from ~1–3 aa/sec (Fig. 4I and

Supplementary Fig. 4B, C). This range is consistent with the range we recently measured using a different reporter without BoxB stem loops[23]. These data therefore demonstrate Ago2 tethering does not impede translation elongation compared to the β-gal control. Moreover, since the Harringtonine-induced runoffs are significantly faster than those in Fig. 4A–G, the data support a model in which Ago2 causes a gradual increase in translation inhibition, leading to ribosome runoff that takes up to 3–4 times longer than that induced by Harringtonine.

**Ago2 tethering mimics miRNA-mediated translational repression at the single-molecule level.** Although previous work provided convincing evidence that Ago2 tethering recapitulates

**Fig. 4 Progressive loss of translation upon Ago2 tethering. A** Track of a TnT biosensor in a cell expressing tetherable Ago2 and TnT components (smFLAG-KDM5B-15xBoxB-24xMS2 mRNA reporter, Cy3-FLAG-Fab, and JF646 HaloTag-MCP). Left, the mRNA track (yellow line) overlaid. Right, crops (11 × 11 pixels$^2$; 130 nm/pixel) show mRNA, translation (Trnl), and Ago2 tethering signals every 100 s. Below, signals over time (Trnl, solid cyan line; Ago2, dashed gray line). Scale bar, 10 μm. **B** Scatter plot showing single-mRNA tethering (β-gal, purple; Ago2, cyan) versus translation (Trnl) for all tracked TnT biosensors (β-gal, 21 tracks, 11 cells; Ago2, 26 tracks, 11 cells). The Spearman Correlation Coefficient was calculated (0.55 for β-gal, $p = 2 \times 10^{-81}$; −0.10 for Ago2, $p = 2 \times 10^{-4}$). **C** Line plots, as in A, showing average signals from TnT biosensor tracks through time. Signals were renormalized to the first four time points. Left, translation (Trnl.) and tethering (Teth.) signals from tetherable β-gal cells (β-gal-teth., dashed gray line; Trnl, solid purple line). Right, translation and tethering signals from tetherable Ago2 cells (Ago2-teth., dashed gray line; Trnl, solid cyan line). Shaded regions, 95% CIs. Red lines show runoff halftime $t_{1/2}$. **D** Piechart of tracked Ago2-tethered TnT biosensors. "Runoff" tracks showed a gradual loss of translation, "Burst" showed bursty translation, "Low/off" had no/low signals, and "Split" had translation and mRNA signals separated. **E** Sample TnT biosensors tracks with gradual ribosomal runoffs (renormalized as in B; every 3rd crop shown with 3-frame rolling average). **F** Sample bursty translation tracks of single TnT biosensors, plotted as in E. **G** Sample long-term silencing track with high Ago2 signals, plotted as in E. **H** Total translation signal from all tracked TnT biosensors in cells expressing tetherable Ago2 or β-gal (Ago2, solid magenta line; β-gal, solid cyan line) and exposed to HT (Time = 0). Shaded regions, 95% CIs. Red line shows runoff halftime $t_{1/2}$. $N = 9$ (Ago2), 9 (β-gal) cells. **I** Box plot of fitted single-cell HT-runoff rates. $N = 9$ cells expressing tetherable Ago2; $N = 9$ cells expressing tetherable β-gal. The boxes (whiskers) show the 25–75% (0–100%) range.

natural miRNA-mediated translational silencing[31–33,35], we wanted to confirm this at the single-molecule level to further validate our TnT biosensor data. To achieve this, we created modified TnT biosensors containing endogenous miRNA response elements (MREs) in place of the tethering cassette. For MREs, we chose a fragment of the 3'UTR of the POLR3G gene, which is predicted by TargetScan[49] to contain just three endogenous MREs targeted by miR-26-5p. To confirm these MREs repress translation in our cells, we first placed them in the 3'UTR of a simpler reporter that encodes sfGFP-H2B (Supplementary Fig. 5A). Cells loaded with the MRE-containing reporter produced ~40% less sfGFP-H2B after 24 h than cells loaded with a reporter containing mutated MREs, indicating that the intact MREs promote miRNA-mediated silencing (Supplementary Fig. 5B).

We then introduced the MRE cassette in place of the BoxB hairpins in our TnT biosensor as well as a similar biosensor that encoded 10 HA instead of 10 FLAG epitopes. In addition, we introduced the mutated MREs into both of these biosensors (Fig. 5A). These miRNA-based biosensors provide a more natural context than the TnT biosensor, but with the drawback that we can no longer track Ago2 association. Nevertheless, the miRNA-based biosensors can still be tracked at the single-molecule level, allowing us to measure translation at individual reporter foci. Assuming our original TnT biosensor recapitulates natural miRNA-mediated silencing, we would predict that our new biosensors containing endogenous MREs would on average have lower translation signals (presumably due to ribosome runoff) than those containing mutated MREs.

To test this, we coloaded cells with one of two combinations of biosensors: either a FLAG miRNA-biosensor with endogenous MREs and an HA miRNA-biosensor with mutated MREs, or the reciprocal pair of biosensors in which FLAG and HA were swapped (Fig. 5A). This allowed us to directly compare endogenous and mutant MRE-containing biosensors in the same cells (Fig. 5B, Supplementary Video 8). We then quantified the translation signals at individual biosensors in live cells. Translation was reduced by 38–57% (across three replicates) in cells containing our FLAG-based biosensors with endogenous MREs relative to cells containing the mutant versions and by 0–54% (across three replicate) in the reciprocal situation with HA-based biosensors (Fig. 5C, Supplementary Fig. 5C). Thus, in agreement with our prediction, endogenous MREs do significantly reduce the translation signals of individual reporter mRNAs. Moreover, the overall 37–57% reduction in translation in all but one replicate is in good agreement with the 40–60% reduction we observed when tethering Ago2 to the original TnT biosensor (Fig. 2E). Last, we tested if MRE-containing biosensors are also recruited to P-bodies, like the original TnT biosensor. As shown in Supplementary Fig. 5D, E, we

did observe a significant fraction of MRE-containing biosensors within P-bodies (~12%; ranging from 10–17%). This fraction is a bit smaller than that we measured with the TnT biosensor (~30%; Fig. 3D). However, this is to be expected considering only three Ago2 molecules are expected to bind the MRE-containing biosensor, compared to 15 Ago2 molecules tethered to the TnT biosensor. Taken together, these data provide additional single-molecule evidence that tethering Ago2 to the TnT biosensor provides a reasonable model for endogenous miRNA-mediated Ago2 translational repression.

## Discussion

The Translation and Tethering (TnT) biosensor reported here allows for direct visualization of specific regulatory factors and quantification of their impact on translation at individual mRNA molecules in living cells. The TnT biosensor innovation builds off of Nascent Chain Tracking technology, adding an independent tethering cassette composed of repeat BoxB stem loops in the 3' UTR of the biosensor. This makes it possible to track individual mRNA molecules and simultaneously monitor their translational status both before and after tethering events are detected. Since tethering is highly specific and the signal amplified, the impact of regulatory factors on translation can be quantified and their actions on timescales ranging from seconds to hours deconvolved. The TnT biosensor therefore brings us one step closer to performing controlled, single-mRNA biochemical reactions in the natural environment of living cells.

To demonstrate the applicability of the TnT biosensor, we used it to deconvolve the complex translational regulatory dynamics of Ago2. There has been a long-term debate about the precise temporal ordering of translational silencing by Ago2[28]. Multiple models have been proposed, some stating sequential translational repression and mRNA decay, others stating decay occurs co-translationally, and still others stating silencing and decay can be uncoupled[50–59]. According to our data, soon after we can detect Ago2 tethering to an mRNA, translation initiation is partially inhibited, leading to a gradual loss of translating ribosomes as they run off the transcript. Collectively, our data support a model in which the inhibition of translation initiation is an early step in the miRNA-mediated silencing pathway, one that can act independently of other steps, such as mRNA decay. Two lines of evidence support this model: first, translational bursts could occur after an mRNA was silenced by Ago2 tethering (Fig. 4F); second, it appeared that nearly all TnT biosensors had 5' and 3' ends according to smiFISH (Supplementary Fig. 3F, G). Together, this evidence suggests an mRNA that has been translationally silenced by Ago2 tethering can remain intact and later be translated. Thus,

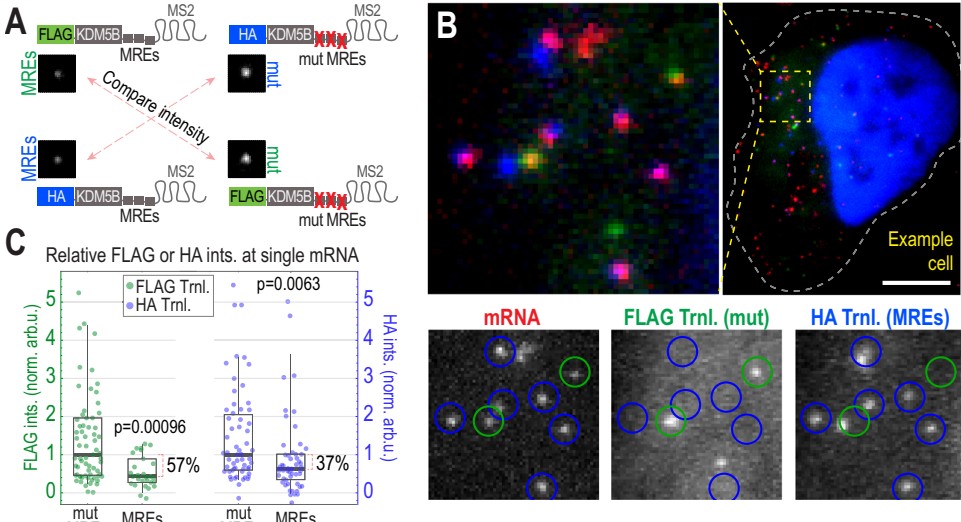

**Fig. 5 miRNA-directed Ago targeting represses translation at the single-mRNA level. A** Schematic of modified TnT biosensors with either endogenous miRNA Response Elements (MREs) or inactive, mutated versions (MUT) inserted in place of the tethering cassette. In each experiment, two modified biosensors (either top two or bottom two) were coexpressed in single cells. To fairly gauge relative translation signals, FLAG (or HA) signals were compared across experiments (red, dashed arrows). **B** An example cell loaded with the top two biosensors in A. Image above shows a whole cell and a representative cytoplasmic patch. Image below shows each separated channel from the patch (intensity auto-adjusted), with detected translating (Trnl) mRNA circled in blue (indicating translation of HA) or green (indicating translation of FLAG). $N = 69$ mut MREs and $N = 30$ MREs for FLAG; $N = 58$ mut MREs and $N = 54$ MREs for HA. Scale bar, 10 μm. **C** Box plots showing the relative intensity (ints.) of the mutated MREs (mut MREs) and endogenous MREs (MREs) containing modified biosensors from 1 of 3 replicate experiments (see Supplementary Fig. 5C for the other two replicate experiments). Each point represents the measured translation signal from a single detected biosensor ($N = 69$ mut MREs and $N = 30$ MREs for FLAG; $N = 58$ mut MREs and $N = 54$ MREs for HA). The boxes (whiskers) show the 25–75% (5–95%) range. P values were calculated using the Mann–Whitney test, two sided.

inhibition of translation initiation can be decoupled from mRNA decay, at least in our tethering system.

By tracking the TnT biosensor over longer timescales, we also discovered both Ago2 and mRNA signals progressively increased with time. The buildup in Ago2 signals began immediately, during ribosome runoff (Fig. 4C, right), while the buildup in mRNA signals occurred over many hours as mRNA and Ago2 molecules coalesced (Fig. 3A, B). The nature of this progressive accumulation of signals is dependent on diverse, multivalent interactions with endogenous miRISC machinery (Supplementary Fig. 3D, E). In particular, a single tethered Ago2 protein can theoretically bind up to three TNRC6B proteins via three distinct binding pockets[39,60]. Likewise, a single TNRC6A protein can bind up to three Ago2 proteins via three distinct binding sites[38]. In turn, TNRC6 proteins contain long, unstructured domains that can recruit other, more downstream miRISC effectors, such as deadenylases, decapping complexes, and translational repression factors like RCK/DDX6[28]. Such diverse, multivalent protein-protein and protein-RNA interactions are hallmarks of phase-separation[16,39,44,45], and minimal Ago2-TNRC6 complexes have in fact been observed to phase separate both in vivo and in vitro[44,56,61,62], leading to increased sequestration of miRNA target mRNAs and deadenylases[39]. It therefore seems likely the TnT biosensor serves as a proxy for these large RNA-protein complexes. The slow and progressive accumulation in mRNA and Ago2 we observed would imply that our TnT biosensor offers a live-cell, single-molecule window into the seeding of phase-separated bodies.

Although we were able to capture and quantify the translational silencing of our TnT biosensor and its ultimate coalescence into P-bodies over time, we had difficulty capturing and quantifying mRNA decay. Despite our best efforts, we were unable to definitively capture a decay event from tracking single mRNA. Although we saw a ~35–63% reduction in the number of mRNA foci in cells expressing tetherable Ago2 compared to controls

(Supplementary Fig. 2C), the clustering of mRNA in P-bodies made it difficult to precisely count mRNA. Unfortunately, once an mRNA entered a P-body, we were unable to resolve its individual dynamics or determine with confidence if it was intact or already decayed. The ratiometric 3' and 5' FISH signals we measured in Supplementary Fig. 3 would suggest some mRNAs in P-bodies are intact, but the two signals could still be on separate molecules. Furthermore, we were reluctant to use P-body total mRNA fluorescence as a proxy for mRNA count because the crowded environment within P-bodies could alter fluorescence in several ways, including the stripping of MS2 coat proteins, intramolecular FRET between tags, or changes in pH. Because of these complications, we can only conclude that at the very most ~35–63% of TnT biosensors were decayed due to Ago2 tethering.

Our experimental design was based off of earlier work from the Filipowicz and Parker labs, which pioneered the use of the Ago2 tethering system to investigate miRNA-directed Ago2 translational silencing[31,35,56,63]. Using the TnT biosensor, we corroborated their finding that tethering Ago2 to a reporter mRNA leads to a global reduction in reporter protein synthesis (Fig. 2B, C and Supplementary Fig. 2B). Furthermore, by comparing the TnT biosensor to a more natural one with endogenous MREs, we demonstrated that Ago2 tethering remains a good model of miRNA-directed translational silencing, even at the single-molecule level (Fig. 5 and Supplementary Fig. 5).

This work is similar to a companion study by Kobayashi and Singer[64] that developed a different miRNA-based reporter to investigate translational silencing by Ago2 with single-molecule precision in situ. Their measurements also indicate Ago2 silences mRNA translation on the 30–40 min timescale, prior to mRNA decay. Although the two reporter systems are similar, a unique advantage of ours is the amplified tethering signal, which allowed us to track the dynamics of Ago2-tethered mRNA for long periods of time in living cells.

Despite these reassuring results, there are three important caveats of our approach that we should point out. First, the artificial tethering of Ago2 to mRNA bypasses the natural inter-action mediated by miRNA base-pairing. This could misorient tethered Ago2 so that it does not behave in a completely natural way. Nonetheless, the consistency of translation silencing and P-body recruitment between the TnT biosensor and the more natural MRE-containing biosensors we tested suggests that teth-ered Ago2 retains its core silencing functionality. Second, our TnT biosensor contains many stem loops that could lead to the formation of 3'-only RNA fragments that accumulate in P-bodies, an issue well documented in yeast[65,66]. Although we did not see any RNA fragments in our two-color smFISH experiments labeling the 3' and 5' ends of the TnT biosensor (Supplementary Fig. 3F, G), this issue should be tested on a case-by-case basis in the future. Third, our tethering cassette was quite large, requiring 15 BoxB stem loops to track tethering for extended periods of time above background. As we had difficulty detecting tethering using a 5x BoxB tethering cassette, we assume the tethered mRNA we detected had somewhere between 5 and 15 tethered Ago2. Such a large multi-protein/RNA complex could interfere with or alter underlying biological processes. For example, the prevalent clustering we observed may not occur with such high probability when just one or two Ago2 proteins are recruited to an endo-genous mRNA target. In the future, it will therefore be important to reduce the number of stem loops required for detection, or perhaps develop a different tethering strategy with a smaller footprint. One promising strategy, for example, would be the use of fluorescently conjugated miRNA rather than EGFP-Ago2. The Walters lab demonstrated individual miRNA can be tracked in living cells[45,67], so in principle their method could be combined with ours to visualize both miRISC and target mRNA dynamics in a more natural setting.

Moving beyond Ago2, the TnT biosensor can now easily be adapted to tether other proteins of interest. Indeed, tethering has frequently been used in the literature to study a wide variety of RNA-binding proteins in diverse biological settings. For example, tethering has been used to investigate nonsense-mediated decay[68], to screen for effects caused by RNA-binding proteins[69], and to study specific protein domains[34,63,70] or spe-cific amino acid modifications[32]. With the TnT biosensor, these studies can be expanded to the single-molecule level, where their impact can be more thoroughly investigated with higher spatio-temporal resolution. At the same time, in light of the presumed relocalization of the TnT biosensor to P-bodies upon Ago2 tethering, we believe our approach will be generally useful in targeting a biosensor to a specific subcellular environment, such as the nucleus, the ER, or mitochondria, where local translation can be studied[1,19]. Finally, the TnT biosensor could be coupled with optogenetic dimerization domains to enable precision tethering with high spatiotemporal control[71–74]. In short, we anticipate the TnT biosensor will be a valuable new tool in the microscopy toolbox to investigate broad protein-mRNA interac-tions with unprecedented spatiotemporal resolution.

## Methods

### Plasmid construction
All expression vectors were designed using SnapGene software. The TnT biosensor construct (smFLAG-KDM5B-15xBoxB-24xMS2) was cloned using the smFLAG-KDM5B-24xMS2 plasmid from ref. [9] and the pRL-5x BoxB construct from ref. [35]. First, using restriction cloning, 5x BoxB repeats were copied from the pRL plasmid with AgeI sites on each end using PCR and inserted into the KDM5B reporter plasmid's 3' UTR at AgeI. Next, two 5x BoxB repeats were isolated from the pRL plasmid and added at the XbaI site, creating the final construct smFLAG-KDM5B-15xBoxB-24xMS2. The orientation of inserted BoxB stem loops was confirmed by Sanger sequencing (Quintara Biosciences). The plasmids were purified via midi-prep (Machery-Nagel) before loading.

The tetherable Ago2 (λN-EGFP-Ago2), non-tetherable Ago2 (EGFP-Ago2), and tetherable β-gal (λN-EGFP-β-gal) constructs were made by first swapping HA for EGFP in the plasmids λN-HA-hAgo2 or λN-HA-β-gal from ref. [35] using isothermal assembly. The EGFP sequence was obtained from the EGFP-hAgo2 plasmid (Addgene plasmid # 21981), which was a gift from Dr. Phil Sharp[75]. Next, the entire open reading frame was inserted into the plasmid backbone from pFN24A HaloTag CMV*d3* Flexi Vector (Promega) so that the ORF was under a CMVd3 minimal expression promoter. Lastly, we verified the plasmid sequence using whole-plasmid sequencing via de novo assembly[76] (Massachusetts General Hospital DNA sequencing core).

The sfGFP-H2B reporters were constructed from the plasmid sfGFP-H2B-C-10 (Addgene 56367, which was a gift from Dr. Michael Davidson) by inserting a UTR sequence at MfeI and HpaI in its 3'UTR. The "MRE" insert was created from a gene block containing the nucleotides 256-709 of the POLR3G 3'UTR, which was predicted to contain three miR-26-5p MREs with 50 nucleotides of endogenous UTR context flanking the first and last MRE. There were also three other MREs predicted: a miR17-5p MRE, a miR383-5p MRE, and a miR302-3p MRE. miR17-5p is ranked 129th and has a very low abundance in U2OS cells, while miR383-5p and miR302-3p are both undetectable in U2OS cells. Thus, only the three miR-26-5p MREs are expected to be relevant in experiments in U2OS cells. The mutant "mut MREs" insert additionally had two nucleotide mutations for each miR-26-5p MRE (sequence and MRE locations obtained from TargetScan[49]). The insert's orientation was verified by Sanger sequencing (Quintara Biosciences). The MRE-containing translation reporters used SpaghettiMonsterHA or SpaghettiMonsterFLAG constructs (smFLAG-KDM5B-24xMS2 or smHA-KDM5B-24xMS2 from ref. [9]). The MRE or mutant POLR3G 3'UTR gene blocks were inserted at AgeI and XmaI sites in the 3'UTR of each FLAG and HA reporter. The insert orientation was verified by Sanger sequencing (Quintara Biosciences).

The plasmid for tetherable Ago2[5XE] (CMVd3 LambdaN-EGFP-Ago2-5XE) was generated by site directed mutagenesis of the LambdaN-EGFP-Ago2 plasmid to swap the amino acid sequence "SAEGSHTSGQS" for "EAEGEHEEGQE". The plasmid EGFP-DCP1A (CMVd3 EGFP-DCP1A) was created by swapping the Ago2 coding sequence with one for DCP1A from the plasmid pT7-EGFP-C1-HsDCP1a, which was a gift from Elisa Izaurralde (Addgene 25030).

The plasmid for tetherable Ago2ΔTRP (CMVd3 LambdaN-EGFP-Ago2ΔTRP) was generated by inserting an Ago2 fragment containing the ΔTRP mutations P590G and R688S at PasI and SmaI sites within the Ago2 sequence of the λN-EGFP-Ago2 plasmid using isothermal assembly.

### Fab/Frankenbody generation and MCP purification
Cy3 α-FLAG Fab were generated and affinity purified as previously described[9]. Briefly, Fab were generated from monoclonal α-FLAG antibodies (Wako, 012-22384 Anti DYKDDDDK mouse IgG2b) using the Pierce Mouse IgG1 Fab and F(ab')2 Preparation Kit (Thermo Fisher). First, α-FLAG antibody were digested into Fab in a Zeba Desalt Spin Column (Thermo Fisher) containing immobilized Ficin undergoing gentle rotation for 3–5 h at 37 °C. Fab were purified from the digest by centrifugation in a NAb Protein A column. Eluted Fab were concentrated to about 1 mg/mL and either conjugated with Cy3 or stored at 4 °C. Cy3 labeling was performed in small batches of 100 μg Fab at a time. The dye was Cy3 *N*-hydroxysuccinimide ester (Invitrogen) dissolved in DMSO and either used immediately or stored at −20 °C. For labeling, 100 μg of Fab was dissolved in a final volume of 100 μL of 100 mM NaHCO3 (pH 8.5) plus 1.33 μl of Cy3 dye. Fabs were labeled for about 2 h at room temperature during gentle rotation and agitation. The Fab were separated from unconjugated dye in an equilibrated PD MiniTrap G-25 desalting column (GE Healthcare). Fab were concentrated in an Amicon Ultra-0.5 Centrifugal Filter Unit (NMWL 10 kDa; Millipore) to about 1 mg ml$^{-1}$. The degree of labeling (*DOL*) was calculated using the following equation:

$$DOL = \frac{\varepsilon_{Fab}}{\varepsilon_{dye}} \times \frac{1}{\frac{A_{280}}{A_{dye}} - CF} \tag{1}$$

where $\varepsilon_{Fab}$ is the extinction coefficient of Fab (70,000 M$^{-1}$ cm$^{-1}$), $\varepsilon_{dye}$ is the extinction coefficient of the dye used for conjugation (150,000 M$^{-1}$ cm$^{-1}$ for Cy3), $A_{280}$ and $A_{dye}$ are the measured absorbances of dye-conjugated Fab fragments at 280 nm and at the peak of the emission spectrum of the dye (570 nm for Cy3), respectively, and *CF* is the correction factor of the dye (the ratio of the absorbances of the dye alone at 280 nm to at the peak; 0.08 for Cy3). If the DOL was <0.8, this protocol was repeated on the same Fab to increase their DOL to ~1. Fab were stored at 4 °C.

MCP was generated as previously described[9]. Briefly, MCP (His-HaloTag-2xMCP) was purified using its histidine tag with Ni-NTA Agarose (Qiagen) following the manufacturer's instructions with minor modifications. The bacteria were lysed in a PBS-based buffer containing a complete set of protease inhibitors (Roche). In a gravity-flow column, binding to the Ni-NTA resin was carried out in the presence of 10 mM imidazole. After washing with 20 and 50 mM imidazole in PBS, the protein was eluted with 300 mM imidazole in PBS. The eluted protein was dialyzed against a HEPES-based buffer (10% glycerol, 25 mM HEPES pH 7.9, 12.5 mM MgCl2, 100 mM KCl, 0.1 mM EDTA, 0.01% NP-40 detergent, and 1 mM DTT) and stored at −80 °C after snap-freezing by liquid nitrogen.

The GFP-tagged α-HA Frankenbody was generated and purified as previously described[77]. Briefly, α-HA Frankenbody was expressed in *E. coli* BL21 (DE3) pLysS cells transformed with pET23b-FB-GFP (Addgene plasmid 129593), induced with IPTG at O.D. ~0.6, and expressed at 18 °C overnight. The protein was purified in

two steps, first with HisTrap HP 5 mL columns (GE Healthcare) and next with a size-exclusion HiLoad Superdex 200 PG column (GE Healthcare) in a HEPES-based buffer (25 mM HEPES pH 7.9, 12.5 mM MgCl₂, 100 mM KCl, 0.1 mM EDTA, 0.01% NP40, 10% glycerol, and 1 mM DTT). Eluted α-HA Frankenbody was flash-frozen in liquid nitrogen and stored at −80 °C. Only α-HA Frankenbody and MCP with 1–2 freeze-thaws were used in experiments.

**Cell culture.** Human U2OS osteosarcoma cells were grown in 10% (v/v) FBS (Atlas) media DMEM (Thermo Scientific) supplemented with 1 mM L-glutamine (Gibco) and 1% (v/v) Pen/Strep (Invitrogen/Gibco) and grown at 37 °C in 5% CO₂. Cell density was maintained between 20–80%. U2OS cells were purchased from ATCC and were authenticated by STR profiling by ATCC and morphological assessments. We also confirmed that the cell line tested negative for mycoplasma contamination.

**Bead loading.** To bead load TnT plasmid and protein components, cells were plated at 70% confluency on 35 mm glass-bottom chambers (MatTek) 1–2 days before imaging. Cells were bead loaded as described previously[78–80]. Briefly, the media was changed to 10% FBS Opti-MEM (Thermo Scientific) before bead loading. The components were mixed together in a total volume of 4–8 µL: ~100 µg/mL of Cy3 α-FLAG Fab, ~33 µg/ml of purified HaloTag-2xMCP protein, plasmids, and 1x PBS if needed to fill to volume. Concentrations of 1.5 or 1.75 µg of DNA were used for the GFP-containing plasmids (either λN-EGFP-Ago2, λN-EGFP-β-gal, or EGFP-Ago2) or the reporter plasmid (pUb-smFLAG-KDM5B -15xBoxB-24xMS2), respectively. After removing the media, the 4 µL mixture of protein and DNA were pipetted on top of the cells, followed by sprinkling a monolayer of ~106 µm glass beads on top of the cells (Sigma Aldrich). The chamber was tapped ~20 times gently, then the media were replaced. After ~2.5 h, cells were washed three times in phenol-free DMEM with 10% FBS and 1 mM L-glutamine. Before imaging, cells were stained with 200 nM JF646 HaloTag ligand for 15 min, followed by three washes in DMEM with 10% FBS and 1 mM L-glutamine. Cells were moved to the microscope stage-top incubator for imaging ~3 h post-bead loading. Cells were selected for imaging if they (1) displayed all imaging components, (2) had at least 5 mRNA, and (3) had suitably low Ago2 or β-gal levels to detect single-mRNA tethering events above background.

**Transfection.** Transfection was performed in the fixed-cell experiments where no protein loading was needed. All transfections were performed with the LTX Lipofectamine with Plus Reagent kit (Thermo Fisher), per the manufacturer's instructions. Briefly, cells were washed and the media was replaced with 1.75 mL Opti-MEM (Thermo Scientific) directly before transfection. The transfection solution included 2.5 µg DNA plasmid, 7.5 µL Plus reagent, and 7.5 µL Lipofectamine and Opti-MEM brought the solution to 250 µL. This solution was incubated for 5–15 min at room temperature before being added to the cell chamber. Cells were incubated in this transfection solution for 2–4 h before the media was changed back to 10% FBS-DMEM. For the TnT biosensor, 1.5 µg smFLAG-KDM5B-15xBoxB-24xMS2 or smFLAG-KDM5B-24xMS2 plasmid plus 1.0 µg of either λN-EGFP-Ago2, λN-EGFP-β-gal, EGFP-Ago2, or λN-EGFP-Ago2-5xE plasmid were used. For sfGFP-H2B reporter assays, 2.5 µg plasmid was used. Cells were selected for imaging if they (1) displayed all imaging components, (2) had at least 5 mRNA, and (3) had suitably low Ago2 or β-gal levels to detect single-mRNA tethering events above background.

**Cell imaging with the confocal microscope.** Fixed-cell images were acquired on an Olympus (IX83) Inverted Spinning Disk Confocal Microscope with a cascade II EMCCD camera. The objectives used were the 40x oil immersion (0.24 µm per pixel) and the 100x oil immersion (0.096 µm per pixel). For smiFISH experiments, the 405 nm, 488 nm, 561 nm, and 647 nm lasers were used. Each location captured a 15 image Z-stack with a step size of 0.35 µm which was max-projected. For the sfGFP-H2B reporter assay, the 405 nm, 488 nm lasers were used. Each location captured a 15 image Z-stack with a step size of 0.35 µm which was max-projected.

**Cell imaging with the HILO microscope.** All live-cell imaging was performed on a custom built widefield fluorescence microscope with a highly inclined thin illumination scheme[81] described previously[9]. Briefly, the microscope equips three solid-state laser lines (488, 561, and 637 nm from Vortran) for excitation, an objective lens (60X, NA 1.49 oil immersion, Olympus), an emission image splitter (T660lpxr, ultra-flat imaging grade, Chroma), and two EMCCD cameras (iXon Ultra 888, Andor). Achromatic doublet lenses with 300 mm focal length (AC254-300-A-ML, Thorlabs) were used to focus images onto the camera chips instead of the regular 180 mm Olympus tube lens to satisfy Nyquist sampling (this lens combination produces 100X images with 130 nm/pixel). The far-red signal of mRNA visualized by JF646 HaloTag-MCP was imaged on one camera, and the red signal of translation visualized by Cy3 α-FLAG Fab and the green signal of GFP-tagged Ago2, β-gal, or α-HA Frankenbody were imaged on the other camera. A high-speed filter wheel (HS-625 HSFW TTL, Finger Lakes Instrumentation) was placed in front of the second camera to minimize the bleed-through between the red and the green signals (593/46 nm BrightLine for the red and 510/42 nm BrightLine for the green, Semrock). The focus was maintained throughout the

experiments with the CRISP Autofocus System (CRISP-890, Applied Scientific Instrumentation). The Z-stack images were taken with a piezoelectric stage (PZU-2150, Applied Scientific Instrumentation). The laser emission, the camera integration, the piezoelectric stage movement, and the emission filter wheel position change were synchronized by an Arduino Mega board (Arduino). Image acquisition was performed using open source Micro-Manager software (1.4.22)[82].

Live cells were placed into a stage-top environmental chamber at 37 °C and 5% CO₂ (Okolab) to equilibrate for at least 30 min before image acquisition. Imaging size, exposure time, and the vertical shift speed was set to 512 × 512 pixels², 53.64 msec, and 1.13 microsec, respectively. This resulted in the imaging rate of 13 Hz (70 msec per image). To capture the whole thickness of the cytoplasm of U2OS cells, 13 Z-stack of step size 500 nm were imaged such that the Z-position changed every 2 images (for the red and the green signals at each stack). This resulted in a maximal cellular imaging rate of 0.5 Hz (2 sec per volume). When needed, delays between volume captures were used to image at lower frame rates. For long-term single-molecule tracking (Fig. 4 and Supplementary Fig. 4), photobleaching was minimized by imaging mRNA (JF646) every 10 s, while imaging tethering and translation (GFP and Cy3) every 100 s. Laser powers for all movies were: 15–20 mW for 637 nm, 9–20 mW for 488 nm, and 5–15 mW for 561 nm with an ND10 neutral density filter at the beam expander.

TetraSpeck Fluorescent Microspheres (100 nm, Thermo Fisher Scientific) mounted on a MatTek chamber were imaged at the end of each imaging session in order to correct for the slight shift in the alignment of the two cameras. These images of beads were used to generate a transformation matrix using either the GeometricTransformation function in Mathematica (Wolfram Research) or the ProjectiveTransform function in scikit-image in Python to correct for offsets in detected particle positions in each channel.

**Co-Immunofluorescence and RNA smiFISH.** Single-Molecule Inexpensive Fluorescent In Situ Hybridization (smiFISH) plus Immunofluoresence (IF) experiments were performed as previously described[46]. Briefly, cells were transfected (as described in the "Transfection" section). After 24 h, cells were washed 3 times in PBS and fixed in 4% paraformaldehyde (Sigma Alrich) PBS for 20 min at 37 °C. Cells were washed then permeabilized in 0.1 mM TritonX (Sigma Alrich) in PBS for 20 min or 70% ethanol at 4 °C overnight.

Next, smiFISH was performed at room temperature unless otherwise specified. The smFISH hybridization and wash buffers were from Biosearch Technologies Stellaris buffers (SMF-HB1-10, SMF-WA1-60, and SMF-WB1-20) and used as stated in the Stellaris protocol for adherent cells. The probe hybridization was performed for ~12 h at 37 °C. The probe set for smFLAG (5') was designed using the open-source R script Oligostan[46], and the probe set for MS2 (3') was copied from ref. [46].

Lastly, immunostaining was performed at room temperature unless otherwise specified. Cells were blocked with 1:4 diluted Blocking One-P (Nacalai Tesque) in PBS for 1 h, then stained with a 1:500 dilution of α-RCK antibody (MBL, PD009) or α-DCP1A antibody (abcam, ab47811) in 1:4 diluted Blocking One-P for 1 h or 6 h. Cells were washed for 30 min four times in 0.1% Tween-20 PBS or 2x SSC buffer, blocked as above for 1 h, then incubated in DyLight 405-AffiniPure F(ab')2 α-Rabbit antibody (Jackson ImmunoResearch, 711-476-152) diluted 1:2000 in 1:4 diluted blocking buffer overnight at 4 °C or 1 h at room temperature. Cells were washed for 30 min four times in 0.1% Tween-20 PBS and mounted in ProLong Diamond AntiFade Mountant (Thermo Fisher). Following these procedures, fixed cells were imaged as described in "Cell imaging with the confocal microscope".

**Immunofluorescence (without smiFISH).** For experiments in Supplementary Fig. 3D, cells were transfected (as described in the "Transfection" section). After 24 h, cells were washed 3 times in PBS and fixed in 4% paraformaldehyde (Sigma Alrich) PBS for 10 min at room temperature. Cells were washed then permeabilized in 0.1% Triton X100 in PBS for 5 min at room temperature. After washing with PBS, cells were stained with a 1:500 dilution of α-RCK antibody (MBL, PD009) in PBS at room temperature for 1 h. Cells were then washed with PBS 3 times for 10 min before staining with DyLight 405-AffiniPure F(ab')2 α-Rabbit antibody (Jackson ImmunoResearch) diluted 1:2000 in PBS for 1 h at room temperature. Cells were then washed with PBS 3 times for 10 min. Cells were fixed in 4% paraformaldehyde-PBS for 10 mins at room temperature. Cells were then washed with PBS twice before mounting in ProLong Diamond AntiFade Mountant (Thermo Fisher). Following these procedures, fixed cells were imaged as described in "Cell imaging with the confocal microscope".

**Live-cell nuclear reporter accumulation assay.** For experiments in Fig. 2B, C and Supplementary Fig. 2A, cells were bead loaded with 0.5 µg of purified Cy3 α-FLAG Fab, 130 nm of purified HaloTag-2xMCP, 1.5 µg of the TnT biosensor plasmid (smFLAG-KDM5B-15xBoxB-24xMS2) and 1.0 µg of either tetherable Ago2 (λN-EGFP-Ago2), tetherable β-gal (λN-EGFP-β-gal), or non-tetherable Ago2 (EGFP-Ago2), as described in the "Bead loading" section. The HaloTag was labeled for 15 min with JF646 HaloTag Ligand 1 h after bead loading, after which cells were moved to the microscope stage-top incubator. At 4 h post-bead loading, the microscope was programmed to visit multiple locations and take one full cell volume (13 Z-planes with a 0.5 µm step size) every 30 min. This continuous imaging was important to identify and exclude cells that died or divided, or moved away.

All videos were max-Z projected. Masks were hand-drawn on the 4 h (early) and 16 h (late) frames to isolate the nucleus and cytoplasm and used to measure the mean intensity of Cy3 α-FLAG Fab in these locations. Reporter accumulation in the nucleus was calculated:

$$Accumulation = \frac{I_{nuc}^{late}}{I_{nuc}^{early}} / \frac{I_{cyto}^{late}}{I_{cyto}^{early}} \quad (2)$$

where $I_{nuc}^{early}$ and $I_{cyto}^{early}$ were the measured intensity of the nucleus or cytoplasm, respectively, at the 4 h time point, while $I_{nuc}^{late}$ and $I_{cyto}^{late}$ were the measured intensity of the nucleus or cytoplasm, respectively, at the 16 h time point. Two of three experimental days were performed blinded.

**Fixed-cell TnT nuclear reporter accumulation assay**. For experiments in (Supplementary Fig. 2B), cells were transfected with the plasmid smFLAG-KDM5B-15xBoxB-24xMS2 and with either tetherable Ago2 (λN-EGFP-Ago2), tetherable β-gal (λN-EGFP-β-gal), or non-tetherable Ago2 (EGFP-Ago2), as described in the "Transfection" section above. After 24 h, cells were fixed in 4% paraformaldehyde-PBS for 20 min at 37 °C then permeabilized in 0.1% Triton X100 for 20 min at 37 °C.

Cells were blocked with 1:4 diluted Blocking One-P (Nacalai Tesque) in PBS for 1 h, then stained with 0.5 µg Cy3 α-FLAG Fab diluted to 100 µL in 1:4 diluted Blocking One-P. Cells were washed for 30 min four times in PBS and imaged directly, as described in the "Cell imaging with the HILO microscope" section. Any images with multiple cells were cropped such that in the end each image showed a single cell, and a mask was hand-drawn around each cell. The mean Fab signal in the masked cell was measured per image. The imaging and analysis of all three experiments were performed blinded.

**Fixed-cell miRNA target site reporter nuclear accumulation assay**. For experiments in (Supplementary Fig. 5B), cells were transfected with 2.5 µg of the plasmid sfGFP-H2B-MRE-POLR3G-3'UTR or sfGFP-H2B-mut-POLR3G-3'UTR, as described in the "Transfection" section above. After 24 h, cells were fixed in 4% paraformaldehyde-PBS for 20 min at 37 °C then stained in 1:2000 diluted Hoechst 33232 (Thermo Fisher) for 10 min.

Image analysis was performed using custom code in Mathematica (Wolfram Research) to detect and measure the fluorescence intensities of all nuclei. Each background subtracted image in the Hoechst channel was binarized (using a sensible threshold) to create a nuclear mask. Using Mathematica's built-in function "ComponentMeasurements," nuclei that crossed the image edge were discarded, and masks were shape- and size-selected so that only whole, single nuclei were detected. The mask was applied to the 488 nm channel of the image and the total sfGFP fluorescence signal of each nucleus was measured. The imaging and analysis of all three experiments were performed blinded. All code is available on Github [https://github.com/Colorado-State-University-Stasevich-Lab/single-molecule-tracking-python].

**MRE-reporter assays**. These experiments (Fig. 5, Supplementary Fig. 5C) used a modified TnT reporter using endogenous MREs instead of BoxB tethering cassette. Using TargetScan[49], we chose a stretch of the POLR3G 3'UTR that had three miR-26-5p sites targeted by miRNA that are abundant in U2OS cells (Jerez et al., 2019). As a control (called the MUT reporter), we mutated two nucleotides in each miR-26-5p site, rendering them untargetable by Ago2 (Mayr et al., 2007) (plasmids are further described above in "Plasmid construction"). Each experiment used two cell chambers. Cells were loaded with the plasmids described below, along with 0.5 µg of Cy3 α-FLAG Fab, 0.5 µg of GFP-tagged α-HA Frankenbody, and 130 nm of HaloTag-2xMCP (further description of loading in the section "Bead loading"). Each replicate experiment used one chamber loaded with 1 µg of smFLAG-KDM5B-MRE-POLR3G-3'UTR-24xMS2 and 1 µg of smHA-KDM5B-mut-POLR3G-3'UTR-24xMS2 and a second chamber loaded with the reversed combination of smHA-KDM5B-MRE-POLR3G-3'UTR-24xMS2 plus 1 µg of smFLAG-KDM5B-mutPOLR3G-3'UTR-24xMS2. Both chambers were then imaged on our custom HILO microscope. For each cell, 20 frames (11-13 Z-stacks with a 0.5 µm vertical step size) were imaged at a 2 frames*s⁻¹ frame rate. Two of the three experiments were performed blinded.

For experiments in (Supplementary Fig. 5D, E), cells were loaded with 1 µg of smFLAG-KDM5B-MRE-POLR3G-3'UTR-24xMS2 plasmid, 1 µg of EGFP-DCP1A plasmid, and 130 nm of HaloTag-2xMCP (further description of loading in the section "Bead loading"). Chambers were then imaged on our custom HILO microscope. For each cell, 120 frames (11-13 Z-stacks with a 0.5 µm vertical step size) were imaged at 6 s intervals. Colocalization of reporter mRNA and EGFP-DCP1A foci (marking P-bodies) was assessed co-movement for at least five consecutive time frames (30 s). This experiment was performed with 2 replicates.

**Puromycin translation inhibition assay**. Puromycin assays (Fig. 1D, Supplementary Fig. 1B, C, and E) were performed as previously described[9]. Briefly, cells were bead loaded with the TnT components along with either tetherable Ago2 (λN-EGFP-Ago2), tetherable β-gal (λN-EGFP-β-gal), or non-tetherable Ago2 (EGFP-Ago2). While on the microscope, cells were treated with 50 µg/mL of puromycin.

Cells were imaged continuously for 5 frames before and up to 60 frames after treatment at a rate of one full cell volume (11-13 Z-planes with a 0.5 µm step size) per 10 or 30 s. All mRNA foci colocalized with a Cy3 α-FLAG Fab translation signal were detected by hand and further categorized by colocalized tethering signal. The count of tethered/untethered, translating mRNA per cell were quantified per frame.

**Single-particle detection, tracking, and intensity measurement**. Image processing was done in the software program Fiji[83] by Z-projecting each time frame of the 3-dimensional movie. These images were then analyzed using custom Mathematica code (Wolfram Research, available on GitHub) to detect single particles in a semi-automated fashion. Briefly, for each image the background was masked by a hand-drawn outline of the cell and each image was binarized using a bandpass filter to highlight particles between 1 and 7 pixels in size (96 nm/pixel for the confocal, 130 nm/pixel for our custom HILO microscope). The mRNA channel was used to detect spots agnostic to tethering or translation status. Mathematica's built-in "ComponentsMeasurements" function was used to select and filter out larger aggregates from single particles and to categorize mRNA into groups based on the presence of detectable particles with signals above background in each channel. When needed, particles were tracked. Briefly, detected mRNA were linked to the closest mRNA in the next time frame, with a maximum step size of 10 pixels and a shortest track length of 5 frames. All detected particles and tracks were confirmed by hand when necessary.

To quantify particle intensities in each channel, intensities were local-background subtracted. This involved three steps using a 15 × 15 pixel crop (130 nm/pixel) of each detected particle: (1) the central signal was calculated as the mean intensity in a centered disc of diameter 3 pixels; (2) the background signal was quantified by measuring the median intensity in four 9-pixel quadrants in the corners of the 15 × 15 pixel crop; (3) The particle's signal intensity in each channel was computed by subtracting the background signal from the central signal. For fixed-cell images using the confocal microscope (Fig. 3C, D and Supplementary Fig. 3), steps 1 and 2 were modified such that the central disc was 5 pixels in diameter to accommodate larger P-bodies (96 nm/pixel), and the background was the mean intensity in an outer ring of diameter 15 pixels and width 2 pixels (96 nm/pixel).

For the long-term, single-molecule tracking experiments in Fig. 4 and Supplementary Fig. 4, intensity measurements were further refined to get the highest possible signal to noise. For this, we used custom Python code to do "best-Z" projections rather than max-Z projections. This was achieved using the positions of particles tracked in 2D to make a set of 3D crops around each detected particle at each time point. Each 3D crop had an XYZ dimension of 11 ×11 x 3 voxels, respectively, where each voxel had an XYZ dimension of 130 nm x 130 nm x 500 nm, respectively. In these 3D crops, the 3 chosen Z-slices were $Z_{best} \pm 1$ such that $Z_{best}$ was the Z-slice having a maximal mean mRNA intensity inside a central disc of diameter 6 pixels (130 nm/pixel). Four steps were then used to quantify the signal intensities in each channel from individual 3D crops: (1) the 3 best-Z slices were max-Z projected (3-frame moving averages of these best-Z projected crops are displayed in Fig. 4 and Supplementary Fig. 4; the only exception is Fig. 4A, where every frame is shown); (2) the central signal was calculated as the mean intensity in a centered disc of diameter 6 pixels (130 nm/pixel); (3) The background signal was calculated as the mean intensity in a surrounding ring of diameter 10 pixels and width 1 pixel (130 nm/pixel); and (4) The signal intensity was computed by subtracting the background signal from the central signal. To aid in this analysis, each step was viewed using Napari[84]. From the measured signal intensities, CSV files were generated and Seaborn[85] was used to create scatter plots and intensity vs. time plots. All Python code for this analysis is available on Github.

**Harringtonine ribosome runoff assay**. To generate Harringtonine (HT) runoff curves, cells were bead loaded with the TnT components along with either tetherable Ago2 (λN-EGFP-Ago2), tetherable β-gal (λN-EGFP-β-gal), or non-tetherable Ago2 (EGFP-Ago2). While on the microscope, cells were treated with 3 µg/mL of Harringtonine (Cayman Chemical) to inhibit translation initiation[48]. As Harringtonine loses potency if left in an aqueous solution at room temperature, for each experiment fresh Harringtonine aliquots were used. For this, 1 mg of Harringtonine crystalline solid that was provided by the manufacturer was diluted and thoroughly vortexed in 333 µl of DMSO to a final concentration of 3 mg/ml. From this, 30 aliquots of 10 µl (30 µg) were made. Individual aliquots were then either immediately used or were used after a single storage of no greater than 1 month at −20 °C. Just before each experiment, 2 µl (6 µg) was taken from a 10 µl aliquot and further diluted and thoroughly vortexed in 500 µl of cell culture media that was directly removed from the 2 ml present in the imaging chamber. This final aliquot was then incubated at 37 °C for no greater than 10 min until it was added back to the imaging chamber during the experiment, making the concentration of HT in the imaging chamber 3 µg/ml. During this whole procedure, the exposure of HT to light was also minimized since HT is light sensitive according to some manufacturers. Cells were continuously imaged for 5 min before and 60 min after addition of HT, at a rate of one full cell volume (13 Z-planes with a 0.5 µm step size) per minute. For photobleach correction, a hand-drawn mask was used to find the mean, background subtracted whole-cell intensity through time (Supplementary Fig. 4B). The single exponential fit of the translation signal decay was

calculated and divided from single-mRNA translation intensities through time. Images were preprocessed and all mRNA in cells were detected as described in the "Single particle detection, tracking, and intensity measurement" section. Intensities of particles in 3D were quantified using best-Z projections rather than max-Z projections. From these, we plotted the total single-molecule translation signals from single cells treated with HT through time. The average of the first three time points was used to normalize single-cell runoff curves so they began at a value of one. Also, to correct for artifactual translation signals that persisted for the entire experiment, the average signal from the 40–60 min imaging interval was measured and subtracted from the runoff curves so they plateaued to zero intensity at the end of the experiment. The first 30 time points of each normalized runoff curve were then fit to a simple phenomenological model:

$$I(v, t) = \frac{1}{2}\left(1 - \tanh\frac{2v\left(t - t_{1/2}\right)}{L}\right) \tag{3}$$

where $L$ is the length of the ORF in our TnT reporter (1885 aa), $v$ is the average elongation rate, and $t_{1/2}$ is defined such that $I(v, t_{1/2}) = 0.5$ (this parameter effectively shifts the runoff curve left or right to allow for the possibility that some cells respond slower than others to the addition of Harringtonine). Note the slope of the fitted runoff curve at $t_{1/2}$ is just $v/L$, which provides a good estimate of the total elongation time. Fitting was performed with Python using the command scipy.optimize.curve_fit from SciPy. All code is available on Github.

**Reporting summary**. Further information on research design is available in the Nature Research Reporting Summary linked to this article.

## Data availability

The raw and processed data and images generated in this study have been deposited in the Figshare database under "Datasets associated with 'Imaging translational control by Argonaute with single-molecule resolution in live cells'" (https://doi.org/10.6084/m9.figshare.c.5395800).

## Code availability

All custom code used in this study are available in Github (https://github.com/Colorado-State-University-Stasevich-Lab/single-molecule-tracking-python).

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

## Acknowledgements
We would like to thank all members of the Stasevich and Montgomery labs for innumerable critical conversations that helped develop and progress this project. We thank Dr. Hotaka Kobayashi and Dr. Rob Singer for sharing their preliminary data. We thank Dr. Ramesh Pillai for generously sharing with us the BoxB and λN plasmids. We thank Dr. Luis Aguilera for graciously supplying his code for bead alignment correction in Python. We thank Dr. O'Neil Wiggan and Nick Flint for helping with assay development for the fixed-cell experiments. The JF646 HaloTag ligand was a generous gift from Dr. Luke Lavis at HHMI Janelia. We thank H. Scherman for purifying HaloTag-MCP. This work was supported by grants from the National Institutes of Health (R35GM119728 to T.J.S and R35GM119775 to T.A.M). C.A.C. was also supported by the National Science Foundation NRT award DGE-1450032.

## Author contributions
C.A.C., G.G., and N.Z. performed experiments. C.A.C., G.G., and T.J.S. analyzed all data. C.A.C., T.J.S., and T.A.M. designed and planned all experiments. T.J.S. and T.M. assisted C.A.C. and G.G. with microscopy and analysis and wrote the relevant methods sections. T.J.S., T.A.M., G.G., and C.A.C. wrote the main manuscript. C.A.C., G.G., T.M., N.Z., T.A.M., and T.J.S. edited the manuscript. T.A.M. and T.J.S. acquired funding.

## Competing interests
The authors have no competing interests.
