## [Peer Review File · Nature Communications]

Title: Imaging translational control by Argonaute with single-molecule resolution in live cellsEditorial Note: This manuscript has been previously reviewed at another journal that is not operating a transparent peer review scheme. This document only contains reviewer comments and rebuttal letters for versions considered at Nature Communications.

REVIEWER COMMENTS

Reviewer #2 (Remarks to the Author):

The authors have nicely addressed all by earlier comments and I am happy to support publication of this exciting manuscript.

Reviewer #3 (Remarks to the Author):

The revised version of this manuscript by Cialek et al has been significantly improved since the original submission. There are a few minor concerns, which once addressed, will render this manuscript suitable for publication.

1. Line 171: Sentence seems incomplete. Perhaps change to "consistent with prior reports^{32,33,35}".
2. Line 223, Fig3 and SFig3: Based on info in Golden et al, the 5XE mutant diminishes AGO2 target association without a "measurable" decrease in miRNA association. Please amend the text accordingly. Also, how does the translation and mRNA abundances of AGO2-5XE tethered constructs compare to the AGO2-WT tethered construct? How does the AGO2, RCK, DCP1A intensity in mRNA colocalized spots compare between the WT and AGO2-5XE samples? This assessment can help resolve questions pertaining to extraneous multivalent interactions with endogenous miRNAs and mRNAs. The 5XE construct at least partially addresses the skewed multivalency question, by potentially minimizing tethered-AGO2 interaction with other endogenous mRNAs. Additional quantitative assessment of "P-body" spots will be needed to better understand the physiological relevance of these foci and address important questions related to multivalency. For instance, what fraction of RCK or DCP1A spots colocalized with AGO2 only or AGO2-mRNA spots? If you did colocalization analysis of RCK and DCP1A in cells with/without (endogenous/additional ones induced by constructs) biosensors, how do they compare? If you stratify the signal intensity of RCK, DCP1A, AGO2 (WT or 5XE) and mRNA within and outside of colocalized areas, how do they compare (similar to Sfig3D,E)? Since it's been shown in Sheu-Gruttadauria et al (Cell 2018) that Trp-binding regions in AGO2 diminishes phase separation of AGO2-TNRC6(fragments), it will be useful for the authors to test the extent of AGO2(delta-Trp binding regions)/mRNA clustering as a control. Moreover, testing an AGO2 mutant that does not bind miRNAs (and consequently other multivalent miRISC machinery and other mRNAs, diminishing P-body localization) will resolve the construct specific multivalency question. These controls will further establish stronger connection to miRISC function.

Reviewer #2 (Remarks to the Author):

The authors have nicely addressed all by earlier comments and I am happy to support publication of this exciting manuscript.

We thank the reviewer for their time and comments.

Reviewer #3 (Remarks to the Author):

The revised version of this manuscript by Cialek et al has been significantly improved since the original submission.

We thank the reviewer for their comments and are glad to hear the reviewer agrees the manuscript is much improved.

There are a few minor concerns, which once addressed, will render this manuscript suitable for publication.

1. Line 171: Sentence seems incomplete. Perhaps change to "consistent with prior reports^{32,33,35}".

We have made this change.

2. Line 223, Fig3 and SFig3: Based on info in Golden et al, the 5XE mutant diminishes AGO2 target association without a "measurable" decrease in miRNA association. Please amend the text accordingly.

We apologize for the confusion here. We now correct this and clarify the text:

'To confirm that P-body localization was not induced by unintended interactions between tetherable Ago2 and endogenous miRNAs targets, we repeated experiments using an Ago2 phosphomimetic mutant (Ago2^{5XE}) that has significantly impaired association with mRNA targets, although it can still interact with miRNA³².

Also, how does the translation and mRNA abundances of AGO2-5XE tethered constructs compare to the AGO2-WT tethered construct? How does the AGO2, RCK, DCP1A intensity in mRNA colocalized spots compare between the WT and AGO2-5XE samples? This assessment can help resolve questions pertaining to extraneous multivalent interactions with endogenous miRNAs and mRNAs.

To address this concern, we have now quantified mRNA, P-body, and Ago2 intensities in Ago2 wild-type and Ago2-5XE mutant experiments.

A. Signals at all P-bodies

B. Signals at all mRNA

Fig. R1: **A.** A scatterplot showing the intensities of Ago2, either tetherable wildtype Ago2 (Ago2-WT, orange) or tetherable mutant Ago2 (Ago2-5XE, blue) and mRNA at all detected P-bodies (all RCK foci; N=565 for Ago2-5XE; N=730 for Ago2-WT; data from 10 cells). **B.** A scatterplot showing the signal intensities of Ago2 and RCK (marker for P-bodies) at all detected mRNA (all MS2-MCP foci; N=2697 for Ago2-5XE; N=2348 for Ago2-WT; data from 10 cells).

We measured mRNA and Ago2 signals at all detected P-bodies (in **A**) as well as RCK and Ago2 signals at all detected mRNA (in **B**). For both the wild-type and mutant Ago2, the distributions were highly overlapping, indicating the 5XE mutant behaves similar to the wild-type Ago2 (Ago2-WT) in terms of its impact on mRNA intensities and subsequent recruitment to P-bodies.

Since the 5XE mutant is unable to associate with target mRNA, this suggests P-body recruitment is not due to unintended interactions with other endogenous mRNAs, but rather is most likely due to natural multivalent interactions that arise between tethered Ago2 and endogenous miRISC machinery.

Note that in these experiments we did not visualize translation since we used up the translation fluorescence channel with the anti-RCK antibody stain. However, as mentioned in the main manuscript (lines 200-201), by the time mRNA enter P-bodies, their translation signals are not detectable.

The 5XE construct at least partially addresses the skewed multivalency question, by potentially minimizing tethered-AGO2 interaction with other endogenous mRNAs. Additional quantitative assessment of “P-body” spots will be needed to better understand the physiological relevance of these foci and address important questions related to multivalency. For instance, what fraction of RCK or DCP1A spots colocalized with AGO2 only or AGO2-mRNA spots?

We have now examined the fraction of colocalized spots for the Ago2-WT and Ago2-5XE mutants, tracking all Ago2 foci in addition to RCK and mRNA foci. We then determined which foci were colocalized with RCK by counting the number of spots that were within 5 pixels (650 nm) of one another. Overall, there was great variation from cell to cell, so we were unable to detect any statistically significant differences between Ago2-WT and Ago2-5XE cells (**Fig. R2**). The only minor difference (that was just shy of significance, defined by $p < 0.05$) was the number of P-bodies per cell, which was slightly decreased in Ago2-5XE cells compared to Ago2-WT cells:

Fig. R2: A. The number of detected RCK (left), Ago2 (middle), and mRNA (right) foci in cells (data from N=10 cells each). The boxplots show the median and quartiles, while the whiskers extend over the full data range.

Similarly, we were unable to detect any statistically significant differences in the percentage of P-bodies that colocalized with the other factors between Ago2-WT and Ago2-5XE cells (**Fig. R3**):

Fig. R3: The % of RCK foci that colocalize with Ago2 (left), mRNA (middle), or Ago2 and mRNA (right). The boxplots show the median and quartiles, while the whiskers extend over the full data range.

Here, the slight increase in the colocalization fractions seen with the Ago2-5XE mutant can be explained by the smaller number of P-bodies per cell (so each has a higher probability of colocalizing with other factors).

Taken together, these data confirm the Ago2-WT and Ago2-5XE mutants behave similarly in our tethering experiments.

If you did colocalization analysis of RCK and DCP1A in cells with/without (endogenous/additional ones induced by constructs) biosensors, how do they compare?

We have now done a side-by-side comparison of Ago2 and RCK signals in RCK-positive foci (P-bodies) in cells with and without our tethering reporter mRNAs:

Fig. R4: A scatterplot showing RCK and Ago2 signal intensities in all detected P-bodies (RCK foci) in cells without a tethering mRNA reporter (blue dots; N=420) or with a tethering mRNA reporter (orange dots; N=730). The main difference is the number of P-bodies per cell (420 vs. 730), with more P-bodies being present in cells with the tethering reporter mRNA present.

Again, there was little difference in the mean and spread of the data, indicating our reporter mRNAs are not altering P-body environments significantly. Interestingly, there were more P-bodies in cells with tethering reporter mRNA. In combination with the slight increase in P-bodies per cell we saw when using the Ago2-WT compared to the Ago2-5XE mutant (**Fig. R2**), these data suggest it is the combination of the Ago2-WT and the tethering reporter mRNA that help seed additional P-bodies in cells.

If you stratify the signal intensity of RCK, DCP1A, AGO2 (WT or 5XE) and mRNA within and outside of colocalized areas, how do they compare (similar to Sfig3D,E)?

We examined all signals inside and outside of the different colocalized areas. Again, there was little difference between the Ago2-WT and Ago2-5XE mutant:

Fig. R5: A comparison of normalized signals from mRNA (top row), RCK (middle row), and Ago2 (bottom row) from cells expressing either the Ago2-5XE mutant (blue dots) or Ago2-WT (orange dots). $N(\text{mRNA}) = 2697$ for Ago2-5XE and 2348 for Ago2-WT; $N(\text{Ago2}) = 1000$ for Ago2-5XE and 859 for Ago2-WT; $N(\text{RCK}) = 565$ for Ago2-5XE and 730 for Ago2-WT.

As can be seen in these plots, there is significant spread and relatively little difference between the Ago2-WT and Ago2-5XE mutant, again suggesting the two behave similarly in our tethering experiments. Given the relatively minor differences we

observed, we are reluctant to include these data in the main manuscript since, as we mention in the Discussion (lines ~449-454), fluorescent signals within crowded environments such as P-bodies can be altered by multiple local factors.

Since it's been shown in Sheu-Gruttadauria et al (Cell 2018) that Trp-binding regions in AGO2 diminishes phase separation of AGO2-TNRC6(fragments), it will be useful for the authors to test the extent of AGO2(delta-Trp binding regions)/mRNA clustering as a control. Moreover, testing an AGO2 mutant that does not bind miRNAs (and consequently other multivalent miRISC machinery and other mRNAs, diminishing P-body localization) will resolve the construct specific multivalency question. These controls will further establish stronger connection to miRISC function.

We agree this is an interesting mutant to test given its inability to associate with miRISC machinery. We therefore decided to clone it into our tethering reporter. As expected, this mutant did not co-localize with P-bodies to any measurable degree:

Fig. R6: A representative fixed cell expressing tetherable Ago2 WT (Top) and tetherable Ago2 Δ Trp (bottom). P-bodies were stained with α -RCK antibodies. Scale bars, 10 μ m.

Importantly, we also did not observe any obvious impact on translation, being able to see strong translation site at 8-12 hours post transfection with our reporters:

Fig. R7: A representative cell expressing tetherable Ago2 Δ Trp and the reporter mRNA (smFLAG-KDM5B-15xBoxB-24xMS2). The image was acquired approximately 8 hours after loading the TnT components (mRNA reporter, Cy3-FLAG-Fab, and JF646 HaloTag MCP). Scale bar, 10 μ m.

This was striking because at these later timepoints nearly all translation sites would be gone with Ago2-WT. We now mention these experiments in the manuscript (line 223-236). However, because of the lack of any observable molecular phenotype, we did not further characterize the mutant.

REVIEWERS' COMMENTS

Reviewer #3 (Remarks to the Author):

The authors have addressed my questions and concerns. This manuscript provides key insights on intracellular AGO2/miRISC function and I fully support its publication.